# Translocation of vaginal microbiota is involved in impairment and protection of uterine health

Jinfeng Wang[1,2,7], Zhanzhan Li[1,2,7], Xiuling Ma[1,7], Lifeng Du[1], Zhen Jia[1,3], Xue Cui[3], Liqun Yu[3], Jing Yang[4], Liwen Xiao[1], Bing Zhang[1], Huimin Fan[1] & Fangqing Zhao [1,2,5,6 ✉]

The vaginal and uterine microbiota play important roles in the health of the female reproductive system. However, the interactions among the microbes in these two niches and their effects on uterine health remain unclear. Here we profile the vaginal and uterine microbial samples of 145 women, and combine with deep mining of public data and animal experiments to characterize the microbial translocation in the female reproductive tract and its role in modulating uterine health. Synchronous variation and increasing convergence of the uterine and vaginal microbiome with advancing age are shown. We also find that transplanting certain strains of vaginal bacteria into the vagina of rats induces or reduces endometritis-like symptoms, and verify the damaging or protective effects of certain vaginal bacteria on endometrium. This study clarifies the interdependent relationship of vaginal bacterial translocation with uterine microecology and endometrial health, which will undoubtedly increase our understanding of female reproductive health.

[1] Beijing Institutes of Life Science, Chinese Academy of Sciences, Beijing, China. [2] University of Chinese Academy of Sciences, Beijing, China. [3] Department of Gynecology, Aviation General Hospital of China Medical University, Beijing, China. [4] Department of Obstetrics and Gynecology, Peking University Third Hospital, Beijing, China. [5] Center for Excellence in Animal Evolution and Genetics, Chinese Academy of Sciences, Kunming, China. [6] Key Laboratory of Systems Biology, Hangzhou Institute for Advanced Study, University of Chinese Academy of Sciences, Hangzhou, China. [7] These authors contributed equally: Jinfeng Wang, Zhanzhan Li, Xiuling Ma. ✉email: zhfq@biols.ac.cn

The microbiota in different niches of the human body may not be as separate from each other as expected[1–3]. Heterologous bacteria may be transmitted and colonize another tissue or organ universally, stimulate inflammation, and increase the risk of diseases such as cancer. A series of examples of microbial cross-colonization have been noted, such as transmission of symbiotic bacteria from other sites to the uterine cavity and fetus in the body of pregnant women[4,5] and the induction of preeclampsia due to translocation of intestinal bacteria to the placenta[6]. Similarly, it was believed that gastric acid prevents effective microbial communication and translocation between the oral cavity and intestine. However, recently, oral bacteria have been determined to frequently migrate to the intestine via the digestive tract[7,8]. Enrichment of specific oral bacteria such as *Fusobacterium nucleatum* was observed in both pancreatic cancer and colorectal cancer[9,10].

Similar to the gastric juice that separates the oral cavity and the intestine, the cervical mucus plug between the uterine cavity and vagina blocks the free passage of microbes. However, the source of uterine microbes and bacterial species exhibiting a high migration rate is under intense discussion. In contrast to the vaginal microbiota[11,12], there was a lack of understanding of the microbial community in the uterine cavity. Fortunately, such knowledge has been rapidly accumulating owing to the increased sensitivity of microbial detection[13–15]. It has been reported that living microbes may be ubiquitous in the female upper reproductive tract, including in the uterus[16,17]. With respect to the microbial taxa largely shared among different parts of the reproductive tract of the same individual, which demonstrate gradual changes from the vagina to the peritoneum, the microbes harbored in the female upper reproductive tract have been speculated to migrate from the lower genital tract or be transferred from the peritoneal fluid[18].

Various *Lactobacillus* species are the dominant bacteria in the vagina of healthy reproductive-age women[19]. The lactic acid they produce maintains the low pH of the vaginal environment, inhibits the growth of harmful bacteria, and maintains the microecology in a relatively balanced state[20]. In abnormal cases, however, the abundance of vaginal *Lactobacillus* may decrease significantly, causing the pH to rise. Elevated pH leads to the proliferation of harmful bacteria such as *Gardnerella* and *Prevotella*, which leads to dysbiosis and an increased risk of various diseases, including bacterial vaginosis (BV) and urinary tract infections (UTIs)[21,22]. Likewise, alteration of the uterine microbiome is closely associated with various intrauterine diseases[23], such as endometriosis, endometrial polyps, and endometrial cancer[16,24,25], and can even affect endometrial receptivity to blastocyst[26]. Therefore, the ecological stability of the vaginal and uterine microbiota plays an important role in the health of the female reproductive system. The uterine cavity and the vagina are physiologically adjacent channels. Hence, theoretically, the bacteria that colonize the vagina have the opportunity to migrate upward to the uterus via the cervix. Although certain studies have speculated that intrauterine infection is caused by vaginal bacteria ascending to the uterine cavity[27], the communication of microbes between these two body sites is still unclear, and the mechanisms underlying the modulation of the microbiota in utero and induction of disease when vaginal bacteria translocate to the upper reproductive system remain obscure.

In this work, we collect both vaginal and uterine samples from healthy women and women with chronic endometritis and conduct data mining of the reproductive tract microbiomes of more than 1000 samples. By combining these results with those of animal experiments, we reveal microbial translocation in the female reproductive tract and elucidate the effect of perturbation of the vaginal microbiota on intrauterine microbiota and reproductive health. These results demonstrate the interdependent effect of the vaginal microbiota and bacterial translocation on uterine microecology and endometrial health, which will increase our understanding of microbial cross-transmission in the female reproductive system as well as their role in modulating the physical health of women.

## Results

We recruited 145 women aged 19–71 years in the first cohort of our study, 106 of whom had experienced at least one abortion and 95 of whom had given birth with different modes of delivery (72 vaginal delivery vs 23 cesarean section) (Fig. 1a). Both uterine and vaginal samples were collected from each subject. The 16S rRNA gene V3–V4 regions were amplified and sequenced successfully from these samples, yielding a total of 48,659,278 PE250 reads. After merging the PE reads into long sequences and trimming out low-quality and chimera sequences, 40,348,849 V3–V4 region sequences were obtained for subsequent analysis, with an average of ~149,000 reads for uterine samples and ~158,000 reads for vaginal samples. We also retrieved the 16S rRNA gene sequencing data of 308 uterine and 653 vaginal samples of women (aged 15–83 years) from six previous projects[16,18,24,28–30]. Together with the new samples obtained in the present study, a total of 1223 samples were included in the meta-analysis.

**Dynamic changes in the uterine and vaginal microbiome during aging.** Given that aging in humans results in extensive changes in physiological functions and metabolism, it is likely that aging is one of the key factors that affect the human microbiota, including that of the reproductive system. We divided the women into six groups according to their age at 10-year intervals. The uterine and vaginal microbiome were profiled to elucidate the microbial transitions during aging and to explore the potential relationship between the aging process and female reproductive tract disorders.

We first observed that the microbial diversities of both the vaginal and uterine microbiome varied with age (Fig. 1b). The trend lines representing the average Shannon diversities were relatively stable in the age groups under 40 but began to fluctuate in the group of women above 40. In the uterine cavity, the youngest women demonstrated the highest alpha diversity, and the diversity decreased slightly with advancing age. However, in the vagina, microbial diversity and evenness increased gradually with age and demonstrated the highest value in the oldest women (Supplementary Fig. 1), which suggests that both the uterine and vaginal microbiome undergo alterations with aging. We further applied a Pearson's correlation test on unbinned data and found that age was significantly correlated with the microbial diversity in both the uterus ($r = -0.173$, $P < 0.01$) and vagina ($r = 0.29$, $P < 0.001$).

To characterize the uterine and vaginal microbiome at different ages, we calculated the Bray-Curtis (BC) distance of the microbial community between individuals in each age group at the OTU level (Fig. 1c). The interindividual community dissimilarities demonstrated a pattern of an initial increase with age ($r = 0.17$, $P < 0.001$ for uterus; $r = 0.28$, $P < 0.001$ for vagina, Pearson's test), followed by a gradual decrease ($r = -0.05$, $P = 0.006$ for uterus, $r = -0.07$, $P < 0.001$ for vagina), although the alteration was less apparent in the uterine cavity than in the vagina. More specifically, young women aged 20 years and younger exhibited the highest interindividual similarity. In contrast, the greatest interindividual difference was observed in women aged 41–60 years. We further calculated the BC distance of microbiome between individuals across different age groups (Fig. 1d). The uterine microbiome gradually deviated from that of young women, and dissimilarity accumulated with increasing age.

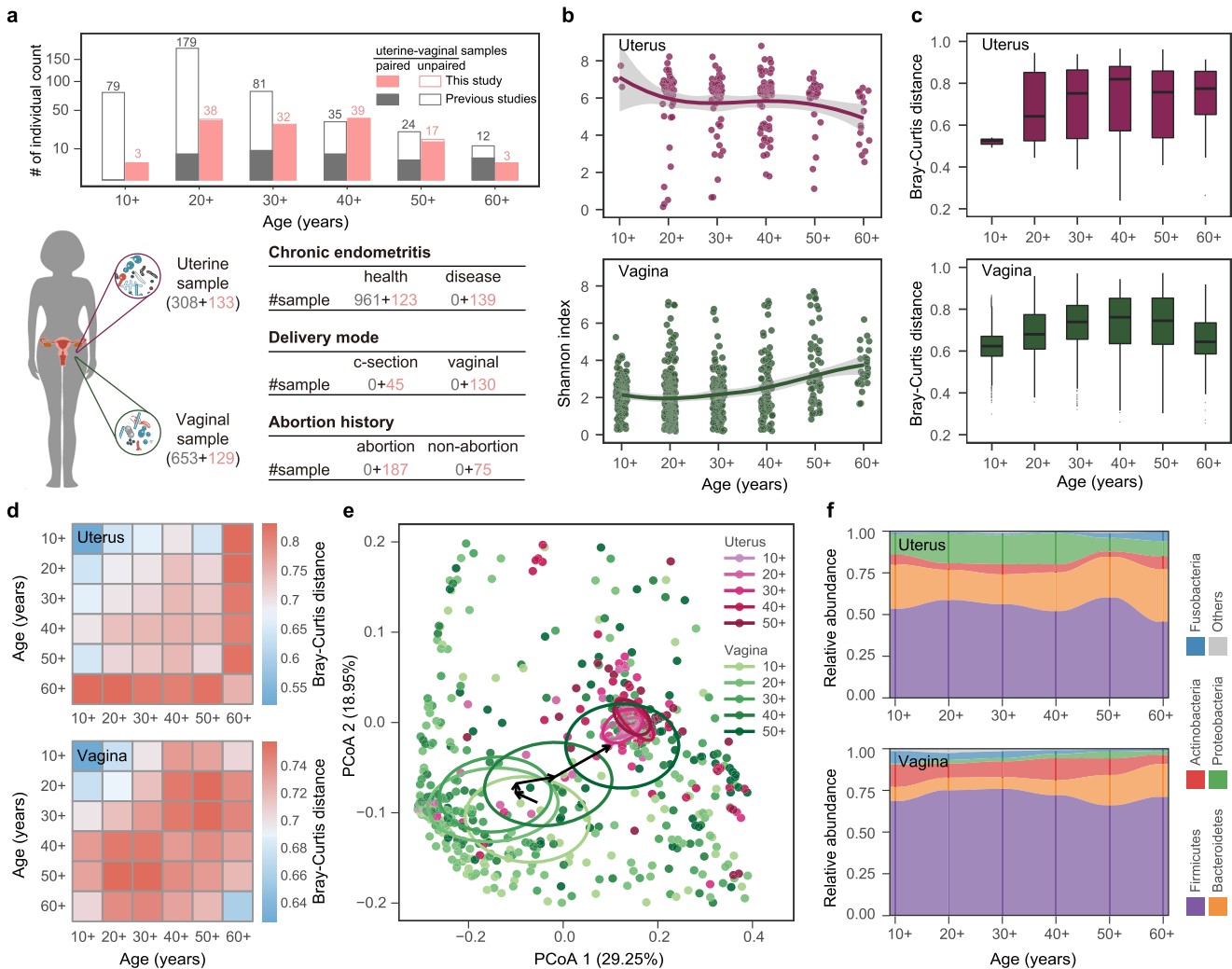

**Fig. 1 Cohort overview and variations in uterine and vaginal microbiome associated with aging. a** The age distribution of women and the number of samples in different groups in this study. Gray bars, red bars, hollow bars, and solid bars represent women from previous studies, from this study, with single uterine or vaginal sample types, and with paired uterine-vaginal sample types, respectively. The number of samples in each age group is given on the bar. The number of sequenced samples from the uterus and vagina and information on chronic endometritis, delivery mode, and abortion history are shown in brackets and tables. Red and gray numbers indicate the samples collected in this study and in previous studies, respectively. **b** Microbial diversities of the uterine and vaginal microbiome in women of different ages. The scales 10+, 20+, 30+, 40+, 50+, and 60+ on the x-axis represent ages <20, 20–29, 30–39, 40–49, 50–59, and ≥60 years, respectively. The shadow around the linear regression trendline shows the 95% confidence interval (CI). **c** The Bray-Curtis (BC) distance of the microbial communities among individuals of the same age group. Box-plot elements are defined as: center line, median; box limits, upper and lower quartiles; whiskers, 1.5× interquartile range; points, outliers. **d** The BC distance of the microbial communities among individuals within the same age group and between different age groups. **e** Principal coordinate analysis (PCoA) of the uterine and vaginal microbiome at each age stage. Each circle shows a 10% CI of each age group, and the arrow connects the vertical center of each circle. **f** The bacterial abundance of the uterine and vaginal microbiome at the phylum level, with different colors correspond to different phyla.

Women with advanced age (over 60 years) exhibited the greatest dissimilarities in the uterine microbiome, compared with women belonging to other age groups. In contrast, variations appeared earlier in the vaginal microbiome, with the largest difference present in the women aged 41–60 years. Both the uterine and vaginal microbiome showed remarkable alterations in the 40+ (40–49) and 50+ (50–59) age groups relative to the neighboring age groups, which is consistent with the results of intragroup alpha and beta diversity, suggesting that the uterine and vaginal microbiota of women in this age range undergo significant perturbation.

We next conducted a principal coordinate analysis (PCoA) based on unweighted UniFrac distance to measure the age-related changes in microbial communities and their clustering relationships. There was a little change in the uterine microbiome. In

contrast, the vaginal microbiome showed gradual changes with advancing age (Fig. 1e). It can be observed from the distribution of vaginal samples that the core regions (center of gravity) of the three younger age groups (10+, 20+ and 30+) were relatively close to each other. In particular, in the 20+ and 30+ year-old samples, the core regions almost overlapped. The vaginal samples of women over 40 years old had a large spatial displacement in the core regions and gradually moved closer to the uterine samples. This is highly consistent with the alpha diversity calculations and the intragroup distances (Fig. 1b, c), suggesting that the age range between 40 and 50 years old may be one of the key stages when the microbiota is more fragile and prone to dysbiosis. For women over the age of 50, the densely distributed area of their vaginal samples overlapped with the uterine samples in PCoA, indicating a more similar community composition.

We attempted to explain the aforementioned changes based on the taxonomic classification. The primary difference between the two body sites was that compared with the vagina, the uterine cavity demonstrated higher amounts of Bacteroides and Proteobacteria and fewer Firmicutes and Actinobacteria. However, the dominant bacterial phyla in the uterine and vaginal microbiome were identical (Fig. 1f). Regardless of body site, the ratio and rank of microbes were relatively stable among women under 40 years old but fluctuated above age 40. Particularly for women over the age of 50, the proportion of Firmicutes and Proteobacteria in the uterine microbiome increased and decreased, respectively, compared with those in the younger age groups. The temporal variation in the vaginal microbiome was accompanied by an increase in the amount of Bacteroides species and a decrease in the amount of Actinobacteria. These variations may account for the alterations in the microbial diversities in both the uterine and vaginal microbiome in this age group, also leading to similarities in the community structures of the two sites.

**Dysbiosis of the uterine and vaginal microbiota associated with chronic endometritis**. To examine the possible impact of age-related changes in the bacterial community on uterine health, we analyzed the uterine and vaginal microbiome in case of intrauterine disease. Among the volunteers recruited in this study, 71 women suffered from chronic endometritis. We compared the differences in microbiome between these patients and women without endometritis (healthy control) to evaluate the correlation between the microbial community and endometrial disease and to explore the intrinsic link between the health of the female reproductive system and age.

The results showed that for women with intrauterine inflammation, both the uterine and vaginal microbiome were significantly different from those of healthy women. First, the microbial diversity of the endometritis and healthy control groups was different (Fig. 2a), with the diversity of the uterine microbiome significantly lower in the endometritis group than in the controls ($P = 0.019$, Wilcoxon test). An opposite trend was observed with respect to the vaginal microbiome; that is, the diversity in the vaginal microbiome was higher in the endometritis group than in the controls ($P = 0.005$, Wilcoxon test), which is consistent with the alterations in alpha diversity associated with age. Second, the community composition was different between the two groups (Fig. 2b). In the PCoA at the OTU level, PC 2 divided samples into two independent clusters ($P = 0.044$, Adonis test), which indicates that endometritis and control women had distinct microbial communities in both the uterine cavity and vagina. While calculating the BC distance between the samples within each group, the distances in the endometritis group were significantly larger than those in the control group ($P < 0.05$, Wilcoxon test), indicating that the microbiome in the endometritis group was highly dissimilar among individuals (Fig. 2c). Likewise, in the vaginal microbiome, the interindividual divergence among the women with endometritis was particularly noticeable and was comparable to the dissimilarities between the endometritis and control groups.

To further illuminate these differences in microbial diversity and community structure, we subsequently grouped samples based on their health status and calculated the relative abundance of the dominant bacteria (Fig. 2d). The proportions of the dominant bacteria in the uterine cavity were similar among controls but fluctuated in women with endometritis. Increases in *Ruminococcus* and *Clostridium* abundance and decreases in *Megamonas* and *Lactobacillus* abundance significantly reduced the similarities of the microbiome among endometritis individuals. A similar phenomenon was also observed in the vaginal

microbiome. *Lactobacillus*, which had an absolute predominance in the vaginal microbiome of most healthy controls, was reduced to less than 50% in more than half of the women with endometritis. With the decrease in *Lactobacillus* abundance, the amounts of other bacteria such as *Prevotella* and *Gardnerella*, increased. The diversity and dissimilarity of the microbial communities among women with endometritis increased significantly (Fig. 2a–c).

To avoid bias due to differences in age distribution, we then chose the individuals with age between 35 and 45 for comparisons, and similar trends on alpha diversities and community structure were observed (Supplementary Fig. 2a, b). Similar results were also found when we used the samples with the same delivery mode (Supplementary Fig. 2c, d). To quantify the distinctions corresponding to health and disease status, we used LEfSe to identify 43 and 51 discriminatory bacterial taxa in the uterine and vaginal microbiome, respectively (Supplementary Fig. 3). Bacteria such as Fusobacteriales, Leptotrichiaceae, *Clostridium*, and *Sneathia* were discriminatory taxa among both the uterine and vaginal microbiome, suggesting that variations in the proportion of specific bacteria associated with health status may take place at these sites simultaneously.

**Synchronous variations in the uterine and vaginal microbiome**. Considering the dysbiosis of both the uterine and vaginal microbiota in women with chronic endometritis, we wanted to understand whether such changes were similar or synchronized across the two sites. We first explored the disease-discriminatory bacteria identified by LEfSe in each sample and observed that the composition and relative abundance of the discriminatory bacteria were quite similar among individuals with the same health status as well as between body sites (Fig. 2e), indicating that the uterine and vaginal microbiota may have undergone similar selective pressure in response to uterine disease status. The rank of highly abundant discriminatory bacteria was almost the same regardless of health status, and the relative abundance of these taxa at both sites was more variable in the endometritis group than in the healthy group (Fig. 2e and Supplementary Fig. 4a). These results indicate the synchronous alteration of the microbiome between the uterus and vagina in women in association with uterine disease. This synchronous relationship was also demonstrated by showing the direction of change with respect to the relative abundance of the discriminatory bacteria between the healthy and endometritis groups (Supplementary Fig. 4b). We investigated whether the direction of their alterations was towards enrichment or depletion in the endometritis group by calculating the difference in the mean relative abundance of each discriminatory taxon between the healthy and endometritis groups. Most of these taxa (37/51) changed in the same direction in both the uterine cavity and vagina in the endometritis group. Taken together, these findings strongly indicated that there is cross-consistency in the relative abundance of specific bacterial taxa and their directions of change associated with chronic endometritis.

To reconstruct the broader relationships among reproductive tract microbes, we calculated the correlation coefficient between the uterine and vaginal microbiome for each OTU (Fig. 2f). Most of the OTUs were negatively correlated in either the healthy group (257/300) or the endometritis group (203/300). Notably, 232 of 300 OTUs maintained the same direction of correlation irrespective of the health status, indicating that the correlation in the relative abundance of any given bacterial taxon was relatively constant between the uterine cavity and vagina. Based on the bacterial abundance, the correlation at the OTU level, and the differences associated with the disease, 7 genera were screened to

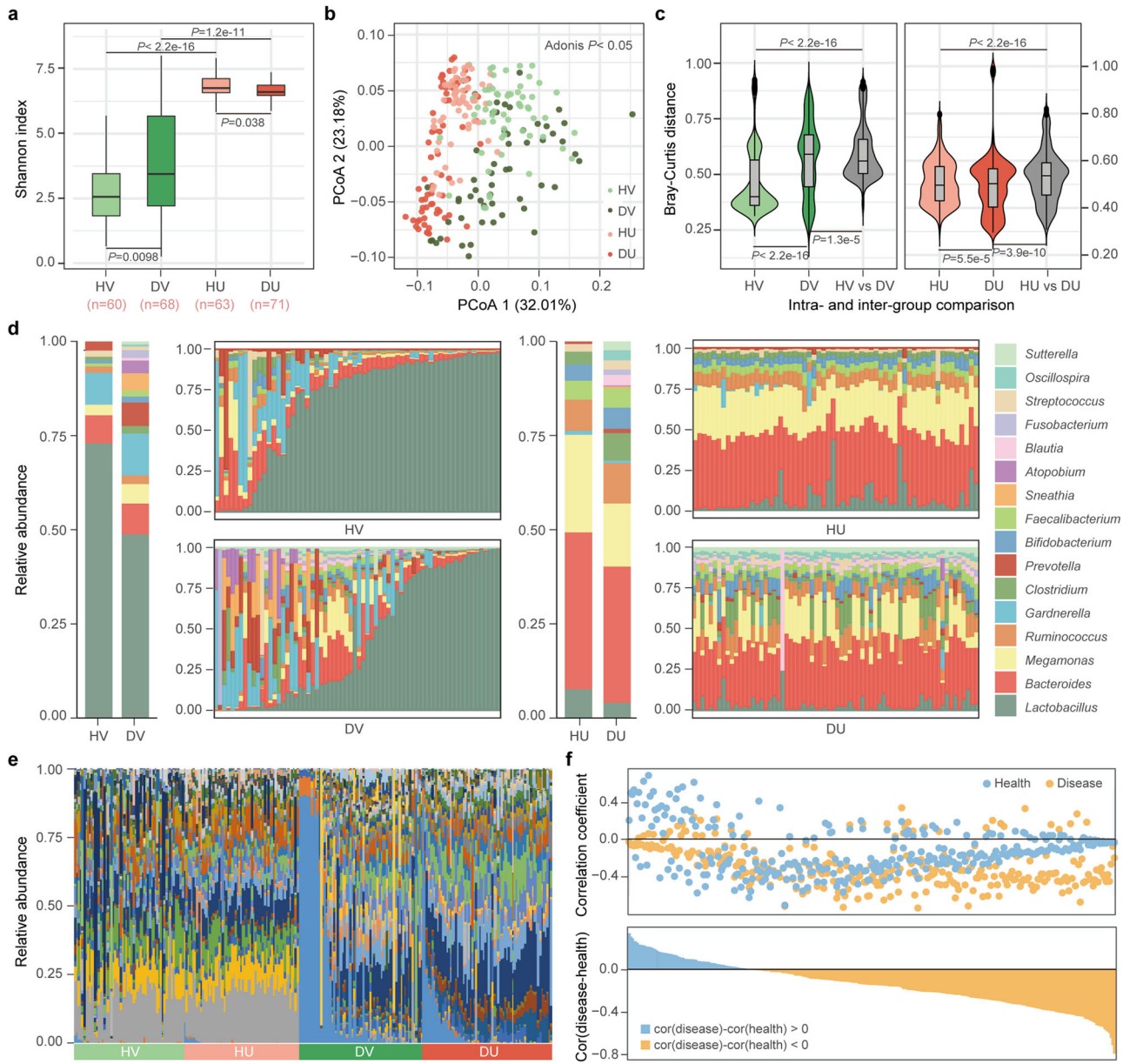

**Fig. 2 The uterine and vaginal microbiome of healthy women and women with chronic endometritis.** HV, DV, HU, and DU represent healthy vagina, endometritis vagina, healthy uterus, and endometritis uterus, respectively. **a** Microbial diversity of the uterine and vaginal microbiome in women with different health statuses. Box-plot elements are defined as: center line, median; box limits, upper and lower quartiles; whiskers, 1.5× interquartile range; points, outliers. P values were determined by two-tailed Wilcoxon test. **b** Principal coordinate analysis (PCoA) of the uterine and vaginal microbiome of different health statuses. The significance of separation of two clusters was measured with the Adonis test (P < 0.05). **c** The Bray-Curtis (BC) distance of the microbial communities among individuals within the same group and between different groups. The violin with box plot shows the median and interquartile range, and the width of the violin represents the density distribution of the BC distance values. P values were determined by two-tailed Wilcoxon test. **d** The relative abundance of the dominant genera in the uterine and vaginal microbiome of different health statuses. The two bars on the left show the average relative abundance of each group. **e** The relative abundances of discriminatory bacteria enriched in women with chronic endometritis. Linear discriminant analysis (LDA) effect size (LEfSe) analysis was performed to identify discriminatory bacteria, with an absolute value of log LDA score >3.0. **f** The co-occurrence of the same bacteria between the uterine and vaginal microbiome. In the top panel, the correlation coefficient of each OTU (each point) between the uterine and vaginal microbiome was calculated in women with (Disease group, yellow point) and without (Health group, blue point) chronic endometritis, respectively. In the bottom panel, the difference in the correlation coefficient of each OTU between the Disease group and the Health group is shown in blue and yellow bars.

analyze the co-occurrence relationship between the same and different bacterial taxa across body sites (Supplementary Fig. 4c). Only the different bacterial genera from different body sites (the pies on both sides of the blue dotted line) were involved in distinct co-occurrence relationships under different health statuses. In contrast, the positive or negative correlation of the

same genus between the uterine and vaginal microbiome was consistent in the control and endometritis groups despite the differences in the strength of the correlation (the pies on the two green dotted lines), which was consistent with the analysis results at the OTU level. Accordingly, the co-occurrence of the same genus across body sites or different genera at the same site was

highly comparable between the endometritis and control groups. The results indicate that the cross-body-site relationship of the same bacteria remained stable or underwent synchronous changes under the endometritis status.

Twenty-one bacterial taxa were identified to be strongly correlated across sites with the correlation coefficient set at $|\rho| > 0.4$ (Supplementary Fig. 5a). Of these bacteria, *Lactobacillus*, *Prevotella*, *Gardnerella*, *Ureaplasma*, and nine other genera were positively correlated. We next classified OTUs with high and similar abundance in the uterine cavity and vagina (Supplementary Fig. 5b). At the genus level (Supplementary Fig. 5c, d), these bacteria showed a large overlap between the healthy (25/30) and endometritis groups (25/35) (Supplementary Fig. 5e), indicating that the bacteria forming a strong correlation across body sites were highly conserved.

We next used source tracking to predict the microbial translocation between the uterus and vagina in the same subject based on 16S rRNA gene sequencing data (Supplementary Fig. 6a). The results showed that a large proportion of uterine microbes originating from the vagina were observed in either endometritis or control women, but the translocation proportion was much more significant in the endometritis group ($P < 0.01$, Wilcoxon test), suggesting that microbes transferred from the vagina to the uterus are correlated with disease status. We further performed metagenomic sequencing on the uterine and vaginal samples of healthy women ($n = 10$) and women with chronic endometritis ($n = 10$) and investigated their strain-level genetic variations by using these metagenomic data (Supplementary Fig. 6b–d). We found that for the same species, a large proportion of identical SNPs were shared between the uterus and vagina, indicating that the microbes of the same species in these two body sites should have the same origin. In addition, nearly all of SNPs identified in the uterus could also be detected in the vagina of the same woman, providing strong evidence that the directionality of translocation of microbiota was from the vagina to the uterus. The observation that the vagina has more unique SNPs may result from the sparsity of microbes in the uterus and the insufficient sequencing coverage of our metagenomic samples.

**Potential factors that may drive the convergence of uterine and vaginal microbiome.** We browsed the medical records of 1612 other women with or without chronic endometritis visiting the gynecology department of one of the two hospitals and observed that age and number of deliveries or abortions were significantly higher ($P < 0.01$, Wilcoxon test) in women with chronic endometritis than in healthy women (Fig. 3a, b). Given that ages and health statuses were associated with the convergence of the microbiome between the two body sites, we next explored the possible impact of clinical factors, pregnancy, and delivery mode on the microbial community of their reproductive system.

The samples were categorized according to various medical factors, and for each category, the BC community distance of the paired samples obtained from the uterine cavity and vagina was calculated. In healthy women <50 years of age, the distance between the uterine and vaginal microbiome increased with age ($r = 0.30$, $P < 0.01$, Pearson's test) and reached its maximum value in the 40–49-year-old group (Fig. 3c). Results similar to those of women 50–59 years old were observed in healthy women with a history of abortion (Fig. 3d) or vaginal delivery (Fig. 3e). The BC distance between their uterine and vaginal microbiome was smaller than that of women who had no history of abortion ($P = 0.07$, Wilcoxon test) or who underwent a cesarean section ($P = 0.03$, Wilcoxon test).

We then analyzed the changes in the abundance of the aforementioned seven dominant genera in different groups and

observed that with increasing age, the trend of the fitted line and the increase/decrease relationship of the abundance of each genus were the same between the uterine cavity and vagina (Fig. 3f and Supplementary Fig. 7a). When using paired uterine and vaginal samples from the same healthy women and testing the bacterial correlations between the two body sites, we found that the correlations of four (*Lactobacillus*, *Bacteroides*, *Gardnerella*, and *Clostridium*) out of seven genera were significant ($P < 0.01$, Pearson's test). Notably, *Lactobacillus*, whose relative abundance was higher in the healthy group than in the endometritis group (Fig. 2d), was depleted in the uterine and/or vaginal microbiome of women over 50 years of age (Fig. 3f). *Clostridium* and *Prevotella*, however, exhibited the opposite trend, with their abundance varying synchronously across body sites, suggesting that they might be closely related to age and the health status of the uterine cavity. Similar results were observed with respect to the variations in the abundance of these genera in women who underwent multiple abortions or vaginal deliveries (Fig. 3g, h and Supplementary Fig. 7b, c). We next measured the biomass of bacteria in vaginal ($n = 40$) and uterine ($n = 40$) flushing fluid by qPCR quantification. As shown in Supplementary Fig. 8, *Prevotella*, *Clostridium*, and *Lactobacillus* in the uterus and vagina were positively or negatively correlated with the age of the women, and each genus exhibited a significant positive correlation between the two body sites ($P < 0.05$, Spearman's test).

**Dysbiosis of uterine microecology triggered by vaginal perturbation.** Since the uterine and vaginal microbiome changed simultaneously or even converged under a variety of clinical contexts, two animal experiments were performed to investigate the possibility of vaginal bacteria passing through the cervical barrier and in utero transplantation, as well as to explore the potential impact of vaginal dysbiosis on the intrauterine microecology. We first exchanged the vaginal microbiota of Brown Norway (BN) and Sprague Dawley (SD) rats ($n = 12$ pairs) three times within 1 week and collected the fluid from lavage of the uterine cavity of each rat 3 weeks later. We subsequently performed amplicon sequencing for the V3–V4 regions of the 16S rRNA genes (Fig. 4a). On average, 155,770 sequences were obtained from each sample. The analysis of microbial diversity at the OTU level showed that the species richness of the uterine microbiome was significantly increased in each rat after exchanging the vaginal microbiota (Fig. 4b). The Shannon diversity index also showed a significant increase ($P < 0.001$, Wilcoxon test) in post-exchange rats (Fig. 4c).

In the PCoA plot based on unweighted UniFrac distance, the uterine samples taken from the rats before and after the exchange of their vaginal microbiota were grouped into two distinct clusters ($P < 0.001$, Adonis test) (Fig. 4d). The distribution of the post-exchange samples was more compact than that before the exchange, indicating that the exchange of the vaginal microbiota could homogenize the uterine microbiota of different individuals. The variations in the diversity and the structure of the uterine microbiome were further elucidated based on taxonomic classification and profiling (Fig. 5e). Before exchange, the uterine cavities of the rats harbored very few bacterial taxa with high abundance except for *Proteobacteria* and *Firmicutes*, and the bacterial taxa varied significantly among individuals, resulting in lower microbial richness and evenness and smaller interindividual similarities. After exchange, the values of the aforementioned parameters increased markedly, demonstrating that the vaginal perturbation could affect the uterine microbiota.

We next collected vaginal lavage fluids from 10 healthy women, 10 women with chronic endometritis, and 10 women

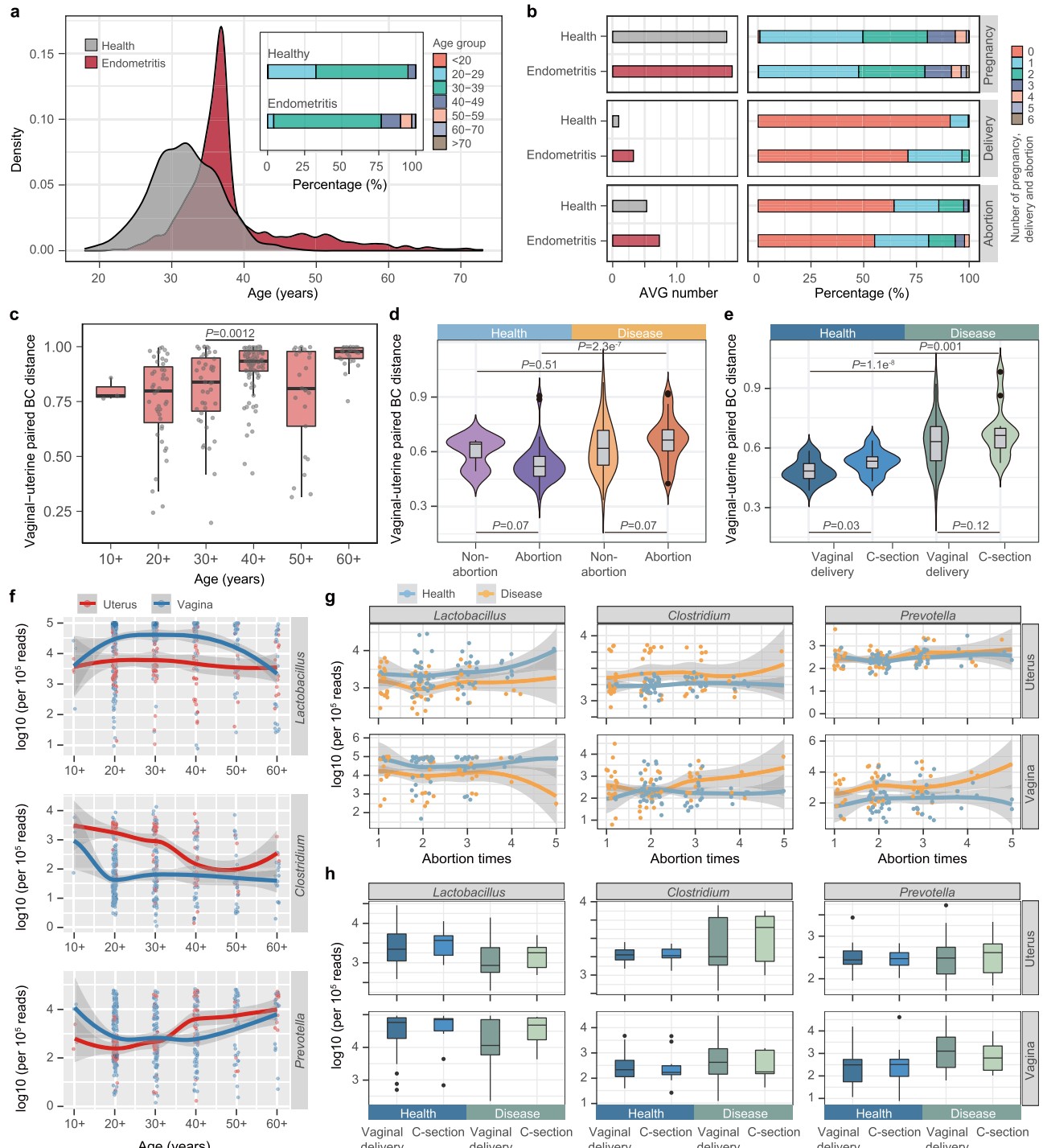

**Fig. 3 Synchronous variations in the uterine and vaginal microbiome and the correlation to clinical factors. a** The age distribution of 1,612 women interviewed from two hospitals. **b** The average (AVG) number and percentage of pregnancies, deliveries, and abortions in women with and without chronic endometritis. **c** Bray-Curtis (BC) community distance of the paired uterine and vaginal samples with age. The scales 10+, 20+, 30+, 40+, 50+, and 60+ on the x-axis represent ages <20, 20–29, 30–39, 40–49, 50–59, and ≥60, respectively. **d** BC distance of uterine-vaginal paired samples associated with abortion history. The distance was calculated at the OTU level in the microbiome of women with and without chronic endometritis. **e** BC community distance of uterine-vaginal paired samples associated with the previous delivery mode. **f** The relative abundance of three bacterial genera in the uterine (red) and vaginal (blue) microbiome with age. The values were normalized to $10^5$ reads in each sample. **g** Relative abundance of three bacterial genera along with the number of abortions. **h** Relative abundance of three bacterial genera along with the previous delivery mode. For **c**–**e** and **h**, box and violin elements are defined as: center line, median; box limits, upper and lower quartiles; whiskers, 1.5× interquartile range; points, outliers; the width of the violin represents the density distribution. P values were determined by two-tailed Wilcoxon test. For **f**, **g** the shadow around the linear regression trendline shows the 95% confidence interval (CI).

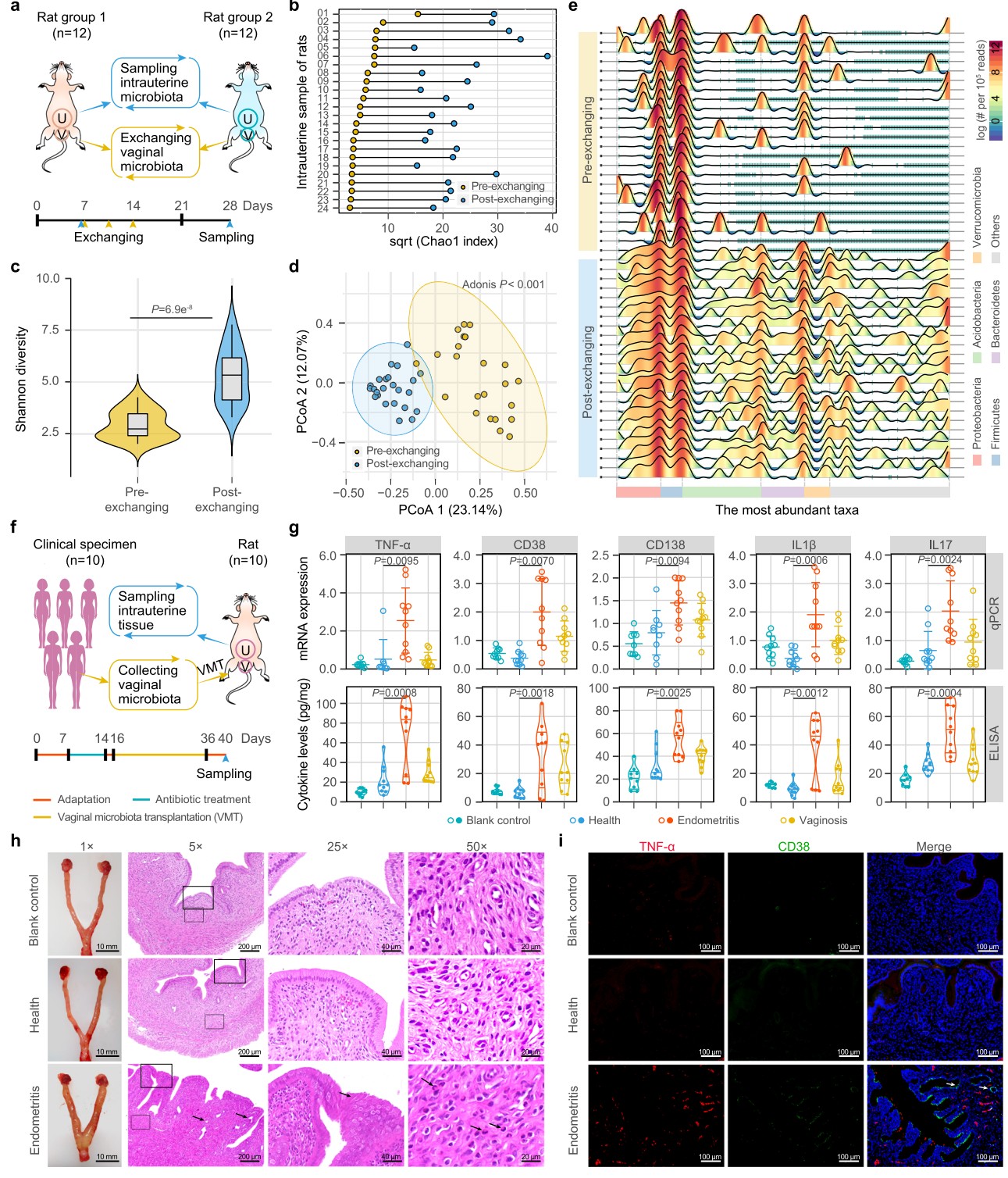

with bacterial vaginosis and transplanted their vaginal lavage fluids into the vaginas of SD rats ($n = 10$) that had previously been submitted to 1 week of antibiotic treatment. After transplantation once a day for 3 weeks, qPCR and ELISA were performed to measure the inflammatory factors, and the changes in uterine morphology and tissue were also examined (Fig. 5f). With respect to the mRNA expression and levels of cytokine and inflammatory factors such as TNF-α, CD38, and IL1β, uterine inflammation was significantly higher in rats transplanted with the vaginal microbiota of women with chronic endometritis ($P <$

0.05, Student's $t$-test) than in those transplanted with microbiota from the healthy group and in the control group (Fig. 5g and Supplementary Fig. 9). Interestingly, the cytokine levels in the bacterial vaginosis group were also elevated but not as high as those in the chronic endometritis group, reflecting the effectiveness of the vaginal microbiota of the latter to stimulate uterine inflammation. The uterine body of the rats in the chronic endometritis group was edematous and enlarged (Fig. 5h) and showed symptoms such as multiple punctate inflammatory lesions (5× field of view), endometrial hyperplasia (25×), and

**Fig. 4 Dysbiosis of the uterine microecology and inflammation of endometrial tissue triggered by vaginal perturbation. a** Study design of one-to-one exchange between the vaginal (V) microbiota of rats. Two groups of female Brown Norway (BN, $n = 12$) and Sprague Dawley (SD, $n = 12$) rats were used as donors and recipients, and the uterine (U) lavage fluid of each rat was sampled 3 weeks after exchange and was used for 16S rRNA gene sequencing. **b** Chao 1 index of the uterine microbiome of each rat before (Pre-) and after (Post-) exchanging the vaginal microbiota. **c** Shannon diversity index of the uterine microbiome in the pre- and post-exchange rats. The violin with box plot shows the median and interquartile range, and the width of the violin represents the density distribution. *P* values were determined by two-tailed Wilcoxon test. **d** Principal coordinate analysis (PCoA) of the uterine samples of the rats before and after vaginal microbiota exchange. The significance of separation of two clusters was measured with the Adonis test ($P < 0.05$). **e** Variations in the diversity and structure of the uterine microbiome in the rats before and after exchange. The taxonomic classification and profiling are shown at the phylum level, with each wave represents a family. Color bar represents the relative abundance of each family per $10^5$ reads. **f** Study design of vaginal microbiota transplantation (VMT) from women to rats. Vaginal lavage fluids were collected from 10 women each in the healthy, chronic endometritis and bacterial vaginosis groups and were transplanted into the vagina of SD rats ($n = 10$) after 1 week of antibiotic treatment. **g** The mRNA expression and cytokine levels of inflammatory factors in the endometrial tissues of VMT rats. For each group, $n = 10$. The upper, center, and lower lines among points show the means ± s.e.m in mRNA expression. The width of the violin represents the density distribution of cytokine levels, and the upper, center and lower lines among points represent upper quartile, median and lower quartile, respectively. *P* values were determined by two-tailed Student's *t*-test. **h** The edematous uterine body and symptoms of the endometrial tissues in the VMT rats. The 25× and 50× fields of view show the areas within the solid and dashed frames of the 5× field of view. The black arrows under 5×, 25×, and 50× fields of view show hyperplastic glands, endometrial epithelium, and inflammatory granulocytes in the uterus of rats transplanted with the vaginal microbiota of women with endometritis, respectively. **i** Immunofluorescence assay illustrating the TNF-α and CD38 signals in the endometrial tissue of the VMT rats. The white arrow shows signals from inflammatory factors. For **h**, **i** images are representative of three independent experiments with similar results.

polynucleosis (50×). The results of the immunofluorescence assay also illustrated a significant presence of TNF-α and CD38 signals in the endometrial tissues of the chronic endometritis group compared to those in the healthy group and the control group (Fig. 5i). These results demonstrate that the vaginal microbiota of women with chronic endometritis can induce inflammation in the uterine cavity, which may be caused by translocated bacteria or inflammatory factors that can cross the cervical barrier and enter the uterine cavity.

**Destructive and protective effects on the endometrium mediated by vaginal bacteria.** To examine the effect of the flux of certain vaginal bacteria on the uterine cavity and its stimulation of uterine inflammation, it was necessary to transplant a single bacterial species into the vagina to build an intrauterine inflammation model. Bacteria such as *Prevotella* and *Clostridium*, which were enriched in the uterine or vaginal microbiome of women with endometritis and demonstrated covariation across body sites were ideal candidates. We thus analyzed the metagenomic sequencing data for the selection of transplanted strains. Similar to the enriched genera identified by 16S rRNA gene analysis, *Prevotella* and *Clostridium* were identified as more abundant in the endometritis group than in the healthy group from the metagenomic data (Supplementary Fig. 10). *Prevotella bivia*, originally isolated from the endometrium (https://www.lgcstandards-atcc.org/products/all/29303.aspx?geo_country=gr), and *Clostridium perfringens*, which has been used previously to construct a model of endometritis[31], were used for the following vaginal transplantation experiment.

We injected *P. bivia* grown to the logarithmic phase into the vaginas of SD rats ($n = 17$) once a day for 3 weeks and then collected samples of intrauterine tissues and uterine and vaginal lavage fluids 3 days after treatment (Fig. 5a). The same transplantation procedure was also performed for *C. perfringens* ($n = 16$). Compared with those in the control group ($n = 10$), the levels of inflammatory factors in the intrauterine tissue, the diameter of the uterine body, and the bacterial biomass of the uterine cavity and vagina all increased significantly in the challenge groups ($P < 0.05$, Student's *t*-test) (Fig. 5b and Supplementary Fig. 11a, b). Significantly positive correlations were observed between these indicators ($P < 0.05$, Student's *t*-test) (Fig. 5c and Supplementary Fig. 11c, d). The uterine body of the rats in the challenge groups was edematous and enlarged (Fig. 5d and Supplementary Fig. 11e). Tissue sections and immunohistochemical staining for TNF-α and CD38 antibodies

showed multiple punctate inflammation and endometrial hyperplasia. Fluorescence in situ hybridization showed that both total bacteria and the transplanted bacterial species specifically exhibited very high biomass in the uterine cavity (Fig. 5d–f and Supplementary Fig. 11e–g). In contrast, these symptoms were absent in the control group. These results indicate that either *P. bivia* or *C. perfringens* introduced into the vagina can ascend to the uterine cavity, induce an inflammatory response in the uterine cavity, and form endometritis-like lesions.

We wondered whether any bacterial species present in the vagina could exert protective effects by reversing the inflammatory effects of *P. bivia* or *C. perfringens*. Given that the relative abundance of *Lactobacillus* in the uterine and vaginal microbiome of healthy women was significantly higher than that of women with endometritis, we selected *Lactobacillus murinus* for the evaluation of its protective effects on the endometrial health of SD rats (Fig. 5g). *C. perfringens* was transplanted into the vagina four times per week for 3 weeks to induce inflammation of the uterine cavity ($n = 15$). In the intervention group, *L. murinus* was transplanted into the vagina once. It was alternately supplemented after each two treatments of *C. perfringens*, such that *C. perfringens* and *L. murinus* were transplanted into the vagina four times and two times per week, respectively ($n = 15$). To test whether the bacterial metabolites in the vagina may also cause uterine inflammation, the transplantation included a group of *C. perfringens* supernatants ($n = 15$). This treatment was similar to that in the *C. perfringens* group, except that the live bacteria were replaced by the supernatant of its culture solution. After 3 weeks of treatment and intervention, the results demonstrated that the inflammatory factors of the intrauterine tissue, the diameter of the uterine body, and the bacterial biomass of the uterine cavity and vagina were significantly higher ($P < 0.05$, Student's *t*-test) in the rats transplanted with only *C. perfringens* than in those belonging to the other three groups (Fig. 5h and Supplementary Fig. 12a). There was no significant difference between the *L. murinus* transplantation group and the *C. perfringens* supernatant group (Supplementary Fig. 12b, c).

We finally performed 16S rRNA gene sequencing on the uterine microbiome of the rats in each transplantation group. PCoA showed that the samples transplanted with *C. perfringens* were significantly different from those of the other groups (Fig. 5i), with more *Clostridium* in the uterine cavity of the former (Fig. 5j). In addition, similar to that in older women and women who experienced multiple abortions, *Bacteroides* and *Prevotella*

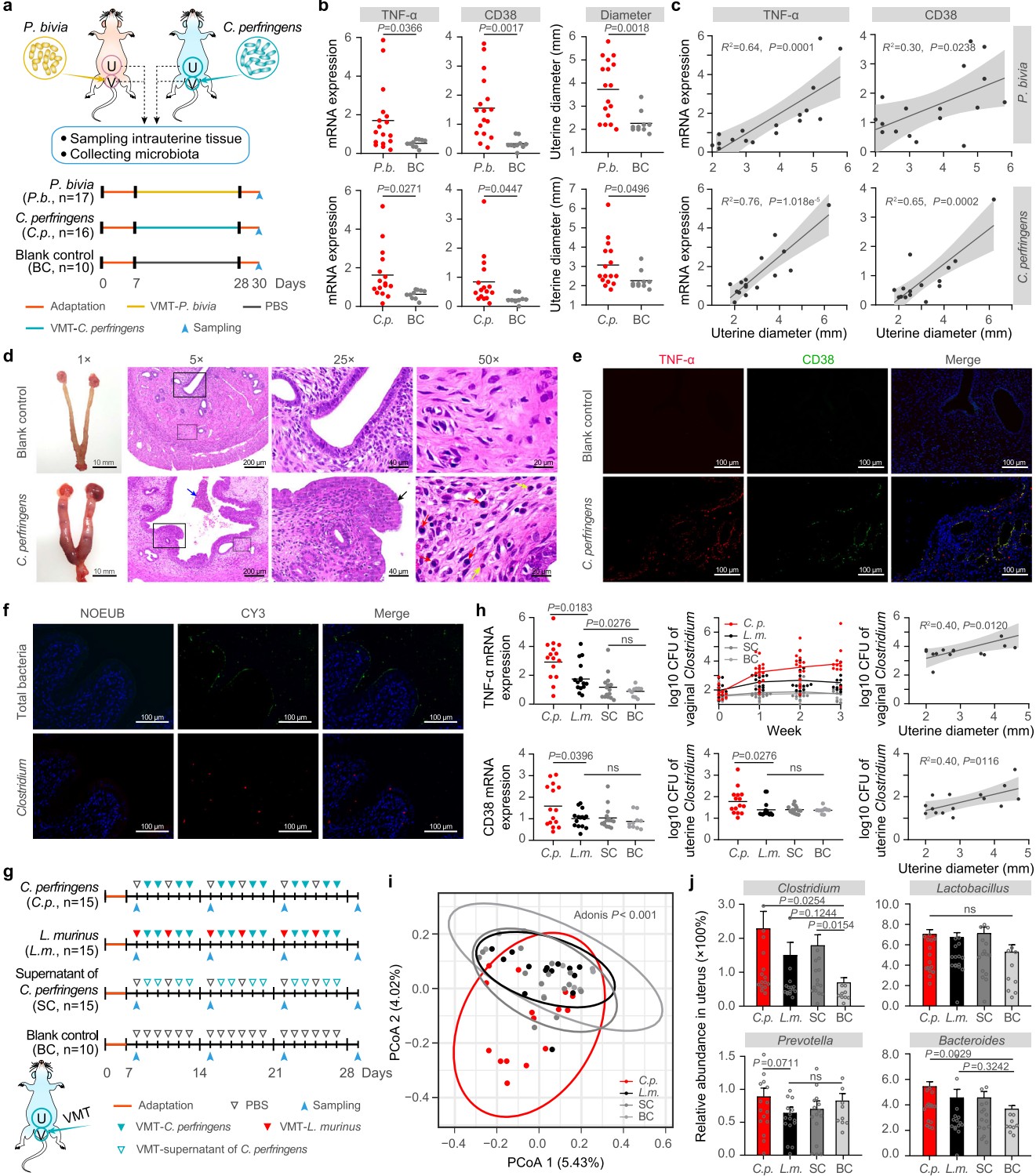

were enriched in the uterine microbiome of rats with vaginal transplantation of *C. perfringens*, whereas *Bifidobacterium* showed the opposite trend (Supplementary Fig. 12d). These findings indicate that bacteria in the vagina can ascend into the uterine cavity to colonize, grow, and cause inflammation, but they do not achieve the same level of stimulation by releasing their metabolites into the vagina only. *L. murinus* can inhibit the growth of vaginal *C. perfringens* and reduce the biomass of uterine *C. perfringens* and some other harmful bacteria and hence may play a protective role in reducing uterine inflammation.

## Discussion

By profiling the uterine and vaginal microbiomes of females of different ages and health statuses, this study adds new evidence indicating that microbes in the uterine cavity inhabit this relatively confined space to form a unique community. It should be noted that contamination from the vagina/cervix cannot be absolutely avoided when collecting uterine samples. In this study, we have used blank and negative controls to exclude potential bacterial contaminations from DNA processing and library preparation. However, more comprehensive external controls from

**Fig. 5 Destructive and protective effects exerted by vaginal bacteria on the endometrium. a** Study design of the vaginal microbiota transplantation (VMT) challenge experiment involving injection of *Prevotella bivia* (*P. bivia*, *P.b.*) or *Clostridium perfringens* (*C. perfringens*, *C.p.*) into the vagina of SD rats. The rats in the blank control (BC) group were injected with PBS instead of bacteria. **b** The mRNA expression of inflammatory factors in the endometrial tissue and the diameter of the uterine body in the *P.b.* ($n = 17$), *C.p.* ($n = 16$) and BC ($n = 10$) rats. *P* values were determined by two-tailed Student's *t*-test, and the lines among points show the means. **c** The correlation between mRNA expression of inflammatory factors and uterine diameter. The shadow around the linear regression trendline shows the 95% confidence interval (CI). **d** The uterine bodies and hematoxylin-eosin staining of the endometrial tissues in the *C. p.* and BC rats. The 25× and 50× fields of view show the areas within the solid and dashed frames of the 5× field of view. The blue arrow under a 5× field of view shows bleeding with blood cells and inflammatory cells (including lymphocyte cells and inflammatory granulocytes) in the uterine cavity. The black arrows under a 25× field of view show hyperplastic endometrial epithelium. The red and yellow arrows under a 50× field of view refer to plasma cells and inflammatory granulocytes, respectively. **e** Immunofluorescence assay illustrating the TNF-α and CD38 signals in the endometrial tissue of the *C.p.* and BC rats. **f** Fluorescence in situ hybridization (FISH) of total bacteria (green) and *Clostridium* (red) in the uterine cavity of the *C.p.* rats. **g** Study design of injecting *C.p.*, *Lactobacillus murinus* (*L. murinus*, *L.m.*), supernatant of *C. perfringens* (SC) and PBS (BC) into the vagina of SD rats. **h** The mRNA expression of inflammatory factors in endometrial tissue, the biomass of vaginal and uterine *Clostridium* and their correlations with uterine diameter in VMT rats. **i** Principal coordinate analysis (PCoA) of the uterine microbiome of the *C.p.*, *L.m.*, SC, and BC rat groups. **j** Relative abundance of *Clostridium*, *Lactobacillus*, *Prevotella*, and *Bacteroides* in the uterine microbiota of the *C.p.* ($n = 15$), *L.m.* ($n = 15$), SC ($n = 15$) and BC ($n = 10$) rat groups. Data are presented as means ± s.e.m. For **d**–**f** images are representative of three independent experiments with similar results. For **c**, **h** linear regression was used to analysis the correlation between two pairs of data, the shadow around the trendline shows the 95% confidence interval (CI). For **h**, **j** *P* values were determined by one-way ANOVA with Tukey's multiple comparison post-hoc test, and data are presented as means ± s.e.m, and ns represents $P < 0.05$.

gloves and air are still needed to exclude all sources of contaminations from sample collection. In terms of community structure, composition, and variation tendency with age, the uterine microbiome is significantly different from that of the vagina, indicating that these observations are unlikely caused by sample contaminations. In addition, the results of our study show that the vaginal and uterine microbiome converge with age. This is a bit of an internal control for the argument that the synchronic variation of the uterine and vaginal microbiome is not an artifact of sample contaminations from vaginal bacteria.

For the first time, we unveiled the synchronic variations of the uterine and vaginal microbiome and observed that age, abortion history, and delivery mode can affect the community structure and similarity between these two body sites in women and uncovered the dysbiosis of the uterine microenvironment caused by vaginal perturbation through animal experiments. Exchange of the vaginal microbiota between rats shifted the structure of their uterine microbiota, and vaginal microbiota transplantation (VMT) from women with chronic endometritis induced inflammation-like lesions in the endometrial tissue of rats. This study also identified candidate bacteria that tend to spread across body sites, which may be responsible for the maintenance or disruption of microbiota homeostasis in the female reproductive system. Their protective or destructive effects on uterine health were confirmed by transplanting bacterial cultures to the vagina and observing cross-site microbial communication, translocation, and colonization from the vagina to the uterine cavity as well as the factors influencing these processes.

Our study indicated that the abundance of variations among certain bacteria was observed in the vagina of women with chronic endometritis. The vaginal microbiota was previously regarded as an indicator of either vaginal or cervical health[21,32]. Similar to the uterine cavity, which is situated deeper into the body, it is difficult to directly monitor the uterine microbial community and in utero health. The findings herein encourage us to explore novel ways for the convenient collection of samples and detection of the vaginal microbiota, which can be used as biomarkers to screen and diagnose asymptomatic uterine diseases or overlooked complications in vitro. Moreover, compared with oral and intestinal bacteria, which may spread into the uterine cavity via the bloodstream under special conditions[5,33], uterine bacteria seem to be more likely to ascend from the adjacent vagina, colonize and lead to adverse pregnancy outcomes or intrauterine diseases. Uterine colonization by BV-related bacteria is not only frequently associated with preterm birth[12,34–36] but

also believed to promote carcinogenesis via microbial-mediated pathophysiological changes[37]. We observed inflammatory lesions in the endometrial tissue of rats transplanted with the vaginal microbiota of BV patients. Combining the results of vaginal transplantation of the microbiota from patients or certain bacterial cultures that caused inflammation in the uterine cavity, our study demonstrated that the translocation of harmful vaginal bacteria can induce inflammation in the uterine cavity. We speculated that the infusion of *Lactobacillus* into the vagina may exhibit a protective function via ecological effects, including the local suppression of the growth of harmful bacteria, decreasing its upward flux, and reducing the stimulation of the endometrium. This inspired us to consider bacterial manipulation as a choice of clinical intervention for the maintenance of uterine microenvironment homeostasis and prevention of disease, similar to the use of vaginal microbiota transplantation for the treatment of intractable BV[38,39].

In this study, less dysbiosis was associated with aging and endometritis in the uterine microbiome than in the vaginal microbiome. One possible reason is that the vagina is a semi-open environment, and its microbiota can be easily influenced. However, the uterine cavity is nearly closed, and hence, its microbiota is more stable and under unique selective pressure. This relative isolation and independence can be disrupted by several factors. For example, we observed that aging, and increase in the number of abortions and vaginal delivery reduced the dissimilarity between the uterine and vaginal microbiome. We speculate that this increase in community similarity may be attributed to the perturbation of the originally confined uterine environment, which promotes bacterial transmission between the upper and lower reproductive tracts. These factors may also lead to increased laxity of the cervix or even cervical incompetence[40], thereby increasing the permeability of the cervix and the likelihood of translocation of vaginal bacteria. In addition, factors such as menopause, hormones or sexual life are also likely to affect the microbiota of the female reproductive system. In this study, we did not gather the information about these factors. However, we discovered that the BC distance of microbiome between age group 60+ and other groups is significantly higher than any other comparisons, which may be contributed to their menopausal or hormonal status.

In women with endometritis, we did not observe a significant increase in the similarity between the uterine and vaginal microbiome in response to abortion history or delivery mode, perhaps since the dysbiosis caused by the disease may mask the

effects of the other factors. Despite epidemiological investigations implying a correlation between reproductive history and endometrial disease[41,42], our study explained the potential connection between the parameters with respect to the reproductive tract microbiota. The results of our study suggested that the uterine niche of the elderly female population gradually deteriorates in women who are older or undergo multiple abortions or vaginal deliveries, and their uterine or vaginal microbiome demonstrate a higher abundance of *Prevotella* and *Clostridium* and a lower abundance of *Lactobacillus*. We also proved in animal experiments that these bacteria inhabiting the vagina can stimulate or inhibit endometrial inflammation. The enrichment and absence of certain bacterial taxa causing endometritis-like changes are characteristics of BV and postpartum vaginal microbiota[22,43]. This might be partially responsible for the clinical linkage of these two diseases with parturition[44,45]. The results of data mining and the experiments in our study may explain the high incidence of endometrial diseases in elderly women, which encourages us to pay attention to the potential threats to the upper reproductive system caused by microbial translocation.

Like all studies that use rodent models to expand physiological and medical understanding, this study also faces the problem of the significant differences in anatomy and morphology between the human and rodent reproductive systems. However, the mouse is still one of the most commonly used models to evaluate the causes and treatments of endometrial or cervical disease[31,46], as the tissue compositions of the human and mouse cervix are very similar and are primarily composed of connective tissue and smooth muscle. Under the action of sex hormones, the variations in the volume and viscosity of cervical mucus were similar between the menstrual cycle in humans and estrus cycle in mice[47]. Rats exhibit a higher degree of cervical compaction than humans and have only a covert estrous cycle rather than overt menstruation, making it more difficult for microbes inhabiting the vagina to pass through the cervix and ascend to the uterus. Hence, in this study, vaginal microbiota transplantation in rats reproducing the uterine inflammation phenotype indicated that similar conditions may also occur in humans. Surprisingly, certain bacterial taxa that were undetected before the exchange of vaginal microbiota were observed in the uterine cavity with high abundance after the exchange in rats. We speculated that the original balance of the uterine microbiota was disrupted, providing opportunities for the colonization and proliferation of less abundant bacteria. In addition, the possibility of the introduction of some exogenous bacteria during transplantation cannot be ruled out. However, in either case, vaginal perturbation will affect the microecological balance of the uterine cavity. In actual situations, the translocation of pathogenic bacteria may not produce as large a biomass in as short a time as in the experimental injection. Nevertheless, it is proposed that there is a way for vaginal bacteria to enter the uterine cavity as well as a perennial and cumulative translocation effect as women age.

## Methods
**Human subjects.** The study was approved by the Ethical Committee of the Aviation General Hospital of China Medical University and performed according to the principles expressed in the Declaration of Helsinki. Female participants were recruited during their visits at the gynecological clinics of the Aviation General Hospital (Supplementary Table 1). Written informed consent was obtained from each participant. Women with abnormal leucorrhea, cervicitis, abnormal levels of sex hormones, TCT- or HPV-positive, malignant tumors, autoimmune disease, severe metabolic diseases, or mental disorders; those who had been administered vaginal medication, antibiotic treatment, or hormonal drugs in the previous 3 months; and those who had engaged in sexual activity or vaginal flushing within 7 days before the hospital visit were excluded from the study. All participants were suspected of having intrauterine lesions but later some of them were ruled out based on hysteroscopy examination by a skilled endoscopist. For hysteroscopy, a 5 mm outer diameter mini-telescope was used, and saline was employed as the

distension medium. The uterine cavity was explored panoramically to identify any surface irregularity. The diagnosis of chronic endometritis was made according to the following criteria: the presence of stromal edema, focal or diffuse periglandular hyperemia, and micropolyps <1 mm in size[48–50]. To identify potential factors that may drive the variation of uterine and vaginal microbiota, clinical data with demographic characteristics, medical and reproductive history, clinical manifestations of the symptoms, laboratory test results, and medication use were collected from women visiting gynecological clinics at Aviation General Hospital or Peking University Third Hospital (Supplementary Table 2).

**Sample collection.** The uterine and vaginal samples were collected during endometrial biopsies from women in the follicular phase of the menstrual cycle. Briefly, vaginal samples were collected as follows: the vagina and the cervix were exposed to a disposable sterile vaginal speculum, and a cotton swab was gently rotated across the vaginal wall for 15 s to absorb vaginal secretions. For intrauterine sampling, an endometrial sampler was gently pushed through the cervical canal along the uterine flexion and into the bottom of the uterine cavity after removing the mucus of the vagina and the cervix using sterile cotton balls. The sampling loop was stretched out of the cannula for complete exposure to the uterine environment, and then the handle of the sampler was rotated 10 times to collect endometrial samples before retracting the sampling loop into the cannula and carefully exiting the cervical canal. During the passage of the sampler through the vagina and the cervix, the sampling loop was kept retracted inside the cannula to avoid contact with the microbes of these two body sites to eliminate the cross-contamination between intrauterine and vaginal samples. The samples were temporarily stored at −20 °C after collection, transferred to −80 °C within 4 h, and stored until further use.

**DNA extraction.** The cotton tips of vaginal swabs and the sampling loops of endometrial samplers were clipped with sterile surgical scissors and vortexed thoroughly to resuspend the samples in 2 mL of PBS (pH = 7.0). The suspensions of each sample were transferred into a centrifuge tube and centrifuged at 12,000 rpm for 5 min. The supernatant was removed, and the precipitates were used for DNA extraction using a QIAamp DNA Mini Kit (Qiagen, Valencia, CA) following the manufacturer's protocol. DNA samples were resuspended in distilled water after final precipitation and elution. DNA quality and concentrations were assessed with the Agilent Bioanalyzer 2100 system (Agilent Technologies, Santa Clara, CA) and a Qubit 3.0 fluorometer (Life Technologies, Waltham, MA), and stored at −80 °C until further use. Ten samples of distilled water were also used in DNA extraction as potential contamination controls.

**16S rRNA gene amplification and sequencing.** Approximately 2–5 μL of extracted DNA and a pair of modified region-specific primers (341F-805R) attached to Illumina paired-end adapters, sequencing primers, and barcodes were used to amplify variable regions 3 and 4 (V3–V4) of the 16S rRNA gene using Q5 Hot Start Polymerase (NEB, Ipswich, MA). In each batch subjected to amplification, negative extraction controls and blank controls were included with distilled water as the PCR template. PCR conditions consisted of an initial denaturation step for 3 min at 95 °C, followed by 25 cycles of denaturation of 30 s at 95 °C, 30 s annealing at 55 °C and 30 s extension at 72 °C, with a final extension for 5 min at 72 °C. The end products were purified using Agencourt AMPure XP beads (Beckman Coulter, Brea, CA), and the quality of purified amplicons was evaluated by electrophoresis on a 2% agarose gel. Positive samples with a bright primary band of approximately 450 bp were pooled in equimolar ratios prior to the generation of 2 × 250 bp paired-end (PE) reads on a HiSeq2500 platform (Illumina, San Diego, CA) with v2 sequencing reagents.

**16S rRNA amplicon sequence analysis.** In addition to the sequencing data generated in this study, the 16S rRNA sequences of 308 uterine and 653 vaginal samples from six previous studies[16,18,24,28–30] were retrieved from the NCBI SRA database (SRP064295, PRJEB14941, PRJEB16013, PRJEB24147, PRJNA481576, PRJNA547595) and used to explore the relationships between uterine and vaginal microbiome and the microbial variation with age in a large population. The samples for these six projects were collected from women aged 15–83 years without chronic endometritis or hyperplasia (Supplementary Table 3). Since the targeted regions of our data and the data retrieved from the databases were V3–V4, V4, or V3–V5 regions, the V4 region of these sequences was uniformly collected for integrated analysis. Briefly, the reference sequences of the 16S rRNA genes were downloaded from the SILVA rRNA database[51], and the V4 regions were marked by mapping the 515F and 806R primers against the reference sequences using Bowtie 2 v2.4.1[52]. Subsequently, we merged the PE reads to form long sequences and mapped them to the reference sequences to capture the sequences falling into the marked V4 regions.

Sequencing data processing and analysis were performed as described previously[53], with modifications. Specifically, a quality filtering step was applied according to the Phred scores via a script split_libraries_fastq.py (-r 3 -p 0.75 -q 3 -n 0) in Quantitative Insights into Microbial Ecology (QIIME) v1.9.1[54]. Chimeric sequences were identified and removed using two QIIME commands, identify_chimeric_seqs.py and filter_fasta. py (usearch61 option that runs the UCHIME algorithm)[55]. High-quality reads were

then merged into long sequences by the overlaps of each pair of PE reads using FLASH v1.2.11 with the implementation of default options[56]. The merged long sequences containing "N" were discarded, and operational taxonomic units (OTUs) were identified from the trimmed sequences with a 97% sequence identity threshold using UCLUST in QIIME. A representative sequence was selected for each OTU, and the Greengenes database was used to generate an OTU table with taxonomy[57]. Samples with observed species less than 50 or Good's coverage lower than 0.8 were discarded from the following analysis.

The relative abundance of each OTU was normalized to the total abundance of all OTUs from each sample. All validated samples were used when analyzing the influence of age on microbial composition. Only the data generated in this study were used to compare the differences of microbiome between women with and without endometritis. Alpha diversity was calculated using the normalized OTU table, followed by rarefaction and Good's coverage evaluation. Both pairwise Bray-Curtis (BC) distance and unweighted UniFrac distance measured at the OTU level were used to determine the dissimilarity of the microbial community, and intergroup comparisons were conducted using Wilcoxon rank-sum tests, Kolmogorov–Smirnov tests, and PERMANOVA in the R package vegan v2.5-6[58]. The phylogenetic tree used to calculate UniFrac distance was created using FastTree v2.1.10[59]. Linear discriminant analysis (LDA) effect size (LEfSe) was conducted to identify discriminatory bacteria, with an absolute value of log LDA score >3.0 considered a differential signature, indicating that the bacteria was significantly able to discriminate between different groups. Pairwise correlation coefficients between the uterine and vaginal bacteria were computed at the OTU and genus level, respectively, using SparCC v0.1.0 with 100 bootstrap replicates to estimate $P$ values[60]. Source tracking of uterine and vaginal microbiome within the same subject was performed based on 16S rRNA taxonomic data by using SourceTracker v1.0[61]. Correlation analysis between the relative abundance of bacteria and the age of women and between the two body sites was performed using Pearson's $r$ test. Benjamini–Hochberg (BH) adjustment was used to control the false discovery rate (FDR) in multiple hypothesis tests.

**Metagenomic sequencing and analysis**. Forty paired uterine-vaginal samples collected from 20 women (chronic endometritis vs control = 1:1) were chosen for whole-genome shotgun sequencing. Metagenomic DNA libraries were constructed using the TruSeq Sample Preparation Kit v2 (Illumina, San Diego, CA, USA) after shearing ~0.1 μg of DNA per sample into ~300 bp fragments using a focused ultrasonicator (Covaris, Woburn, MA). Insert size and concentrations of each library were confirmed using a Fragment Analyzer (Advanced Analytical Technologies, Ames, IA). Libraries were pooled by mixing an equal quantity of each library and sequenced on the HiSeq2500 platform with a paired end (PE) 150 bp sequencing strategy. Raw reads were filtered using the FASTQ quality filter in the FASTX-toolkit v0.0.14 (http://hannonlab.cshl.edu/fastx_toolkit) with the parameters -p 90 -q 25 -Q33. High-quality reads were initially aligned to the hg38 release of the human reference genome using the BWA v0.7.17 algorithm with default parameters[62]. Both members of a read pair were discarded if one read matched the human genome.

The metagenomic read pairs were used for strain-level source tracking as follows: Uterus-vagina paired samples from the same subject were aligned to MetaPhlAn v2.7.6[63] species-specific gene markers to measure microbes shared by two body sites at the strain level. The "mpileup" feature in SAMtools v0.1.19[64] was used to calculate coverage and determine variable sites for each marker gene. For a specific gene marker, the same variation in paired samples of the same woman was considered to be an identical SNP, which was assessed by >40% coverage of this gene marker. If all the identical SNPs from one body site are observed in its paired body site of the same woman, it will be considered as the same strain shared by both uterus and vagina in this subject.

The metagenomic read pairs were also aligned to a custom database for the selection of transplanted strains used in animal experiments. This database consisted of the reference genomes of *Clostridium* and *Prevotella* strains deposited in the Human Microbiome Project (HMP). Only the read pairs that were uniquely mapped to one strain genome with a mapping quality of ≥20 were counted using a custom script.

**Animal experiments**. Animal experiments were approved by the Experimental Animal Welfare Ethics Committee of Institutes of Zoology, Chinese Academy of Sciences, and have complied with all relevant ethical regulations for animal testing and research. Nine-month-old female SPF rats (SPF Biotechnology, Beijing, China) were housed in groups of 2–3 animals per cage in a pathogen-free facility with a 12-h light:dark cycle and ad libitum access to food and water. All rats demonstrated an environmental adaptation period of 7 days before the experiment. For sampling the vaginal or uterine microbiota, 120 μL PBS (pH = 7.0) was injected into the vagina or uterine cavity of each rat, and the lavage fluid was harvested after gentle rinsing three times and stored at −80 °C. Rats were anesthetized by IV pentobarbital prior to the collection of vaginal and intrauterine tissues.

Two groups of female Brown Norway (BN; $n = 12$) and Sprague Dawley (SD; $n = 12$) rats were used as donors and recipients to evaluate the effect of exchanging vaginal bacteria on the uterine microbiota. The exchange procedure was performed three times over 7 days. Briefly, the vagina of one rat was rinsed with 120 μL of PBS, and the washing solution was then injected into the vagina of another rat to exchange the

vaginal bacteria between rat pairs. The rats were kept flat on their backs throughout the procedure to prevent fluid from flowing into the uterine cavity. The animals were adapted for 14 days before sampling their intrauterine microbiota.

Vaginal washing solution obtained from clinical patients with endometritis ($n = 10$), patients with vaginosis ($n = 10$), and healthy women ($n = 10$) was transplanted to SD rats. The clinical specimens were centrifuged at 1000 rpm for 5 min to remove the visible residue, and the supernatant was harvested and mixed together. The mixture was centrifuged at 6000 rpm for 5 min to harvest the bacteria. The precipitate was resuspended in 120 μL sterile and anaerobic PBS (pH = 7.0) before vaginal microbial transplantation (VMT), with PBS as the blank control. After adaptation for 7 days, SD rats ($n = 10$) were treated with a combination of antibiotics (ABX) containing 100 μg mL$^{-1}$ neomycin, 100 μg mL$^{-1}$ penicillin, 50 μg mL$^{-1}$ vancomycin, and 100 μg mL$^{-1}$ metronidazole (Sigma, St. Louis, MO) to reduce the amount of endogenous bacteria before they were recolonized by clinical transfers. Following antibiotic treatment, the recipient rats received clinical vaginal bacteria via VMT daily for 21 consecutive days. Their intrauterine tissues were collected after an adaptation period of 3 days.

Strains *Prevotella bivia* ATCC29303, *Clostridium perfringens* BNCC185933, and *Lactobacillus murinus* C-30 were cultivated anaerobically at 37 °C in Gifu anaerobic medium (GAM). Two independent experiments were conducted. In the first experiment, approximately $1 \times 10^8$ CFU of bacteria cultured in the logarithmic phase in 120 μL GAM were administered to three rat groups: *P. bivia* ($n = 17$), *C. perfringens* ($n = 16$), and blank control ($n = 10$). The recipient rats received the cultured bacteria via VMT daily for 21 consecutive days. The vaginal and uterine washing solution and intrauterine tissues were collected after adaptation. In the second experiment, the culture medium supernatant after the elimination of *C. perfringens* by centrifugation at $13,000 \times g$ for 20 min was used as a negative control, and its sterility was checked by culturing the suspension to examine the absence of the growth of bacterial clones. The rats were divided into four groups: *C. perfringens* ($n = 15$), *L. murinus* ($n = 15$), *C. perfringens* supernatant ($n = 15$), and the blank control ($n = 10$). In each week, the rats in the first group received $1 \times 10^8$ freshly cultured *C. perfringens* on the second, third, fifth, and sixth days via VMT and received PBS (pH = 7.0) on the first and the fourth days. The experiment was repeated for 3 weeks. The treatment course of the *L. murinus* group and the supernatant of the *C. perfringens* group was the same as that of the first group, except that PBS was replaced with *L. murinus* and *C. perfringens* was replaced with its supernatant. PBS was used in the blank control group at all the same time points as above. The washing solutions from the vagina and the uterine cavity and intrauterine tissues were sampled on the first, eighth, fifteenth, and twenty-ninth days.

**Morphological and histological examinations**. Once collected, the uterine tissues of the rats were photographed immediately, and the uterine diameter was measured using a Vernier caliper. The uterus was split into two parts: one part was stored at −80 °C, and the other part was fixed in 4% paraformaldehyde (PFA) solution for 24 h and embedded in paraffin. Approximately 5-μm-thick sections of the uterine tissue were stained with hematoxylin-eosin. Whole-tissue slide scans were examined at ×40 magnification under a microscope slide scanner (Pannoramic MIDI, 3DHISTECH), and ×10, ×25, and ×50 images were analyzed with CaseViewer_2.3 image analysis software.

**Quantitative PCR and gene expression analysis**. To determine the absolute amount of bacteria at the genus level in the uterine cavity and vagina of women, a universal 16S RNA primer pair (1369F; 1492R) for total bacteria and three primer pairs specific for *Prevotella* (F: 5′-CCAGCCAAGTAGCGTGCA-3′; R: TGGGACC TTCCGTATTACCGC-3′)[65], *Clostridium* (SJ-F: 5′-CGGTGAAATGCGTAGA KATTA-3′; SJ-R: 5′-CGAATTAAACCACATGCTCCG-3′)[66] and *Lactobacillus* (LbLMA1-rev:5′-CTCAAAACTAAACAAAGTTTC-3′; R16-1: 5′-CTTGTACA CACCGCCCGTCA-3′)[67] were used (Supplementary Table 4). Real-time qPCR analysis was performed using SYBR Green PCR master mix (Yeasen, Shanghai, China) on an ABI 7500 real-time PCR system (Applied Biosystems, Darmstadt, Germany).

Total RNA was extracted from the uterine tissue of rats using TRIzol reagent (Invitrogen, Carlsbad, CA) according to the manufacturer's instructions to measure the gene expression of inflammatory factors in the endometrial tissues. Approximately 1 μg RNA was used as a template for the reverse transcription reaction to synthesize complementary DNA (cDNA) with reverse transcription enzyme (Yeasen, Shanghai, China). The primers used are listed in Supplementary Table 4. The results of gene expression are presented as a percentage expression of each gene normalized with glyceraldehyde-3-phosphate dehydrogenase (GAPDH) as a reference.

**Enzyme-linked immunosorbent assay**. Total protein was extracted from the uterine tissues of rats using RIPA lysis buffer (Beyotime, Shanghai, China). The resulting suspension was centrifuged at $2000 \times g$ for 30 min, and the supernatants were harvested. The levels of inflammatory mediators, including TNF-α, IL1β, IL-17, CXCL5, CD138, and CD38, were determined using a MILLIPLEX MAP Rat Cytokine/ Chemokine Panel (Millipore, Billerica, MA) on a Luminex 100 system (Luminex, Austin, TX). All assays were conducted according to the manufacturers' guidelines.

**Immunofluorescence**. Immunofluorescence (IF) was performed using anti-TNF-α (ab109322, Abcam, Cambridge, MA) and anti-CD38 (bs-0979R, Bioss Antibodies, Beijing, China) antibodies. After deparaffinizing and rehydrating, the formalin-fixed or paraffin-embedded sections were transferred into a microwave in a 10 mM citrate antigen repair solution (Servicebio Biotechnology, Wuhan, China) for antigen retrieval. The sections were then blocked in 3% BSA (Sigma, St Louis, MO) for 1 h, followed by overnight incubation in the primary antibody dissolved in 3% BSA (anti-TNF-α: 1/50; anti-CD38: 1/100). Detection and labeling were performed using secondary antibodies conjugated to Alexa Fluor-488 donkey anti-rabbit IgG (H + L) (1/500, A21206, Invitrogen, Carlsbad, CA) or Alexa Fluor-594 goat anti-rabbit IgG (H + L), F(ab')2 Fragment (1/500, 8889S, Cell Signaling Technology, Beverly, MA) fluorophores, and imaging was performed as described above.

**Fluorescence in situ hybridization**. Fluorescence in situ hybridization (FISH) was performed as described by Choi et al.[68]. Briefly, formalin-fixed or paraffin-embedded sections were subjected to deparaffinization and rehydration. Specific probes for FISH, including the Eubacterial probe (EUB338 I-III) mixture, were used to examine total bacteria. The NON338 probe served as nonspecific control. The *Clostridium*-specific probe served as a genus-specific probe complementary to the partial 16S rRNA region of *C. perfringens* BNCC185933. The *Prevotella*-specific probe served as a genus-specific probe complementary to the partial 16S rRNA region of *P. bivia* strain ATCC29303 (Supplementary Table 5). All probes were 5′-labeled with digoxigenin, and anti-digoxigenin/horseradish peroxidase antibodies were used as secondary antibodies.

**Microbial sequencing and quantification in rats[50]**. DNA was extracted from the uterine and vaginal washing fluid of rats using a QIAamp DNA Mini Kit (Qiagen, Valencia, CA) according to the manufacturer's instructions. 16S rRNA gene amplification, sequencing, and data analysis were performed as described above. A 10-fold serial dilution of the DNA templates ($10^0$–$10^8$) was used to construct the standard curve, total bacteria was chosen as the endogenous control, and the relative abundance of target bacteria was calculated using the $2^{-\Delta\Delta Ct}$ method.

**Statistical analysis of the animal experiments**. The statistical analysis of the animal experiments was performed using GraphPad Prism v8.0 software (Graph-Pad, San Diego, CA). $P$ values less than 0.05 as determined by unpaired two-tailed Student's $t$-test or ANOVA were considered to indicate statistically significant differences. Correlation analyses were performed based on Spearman's rho statistic. Rate comparisons were performed with Pearson's $\chi^2$ test or Fisher's exact test.

**Reporting summary**. Further information on research design is available in the Nature Research Reporting Summary linked to this article.

## Data availability

The sequencing data generated in this study have been deposited in the NCBI SRA database under accession number (PRJNA737052). The public 16S rRNA sequences used in this study are available in the NCBI SRA database under accession number SRP064295, PRJEB14941, PRJEB16013, PRJEB24147, PRJNA481576, and PRJNA547595. The reference sequences of the 16S rRNA genes are available in the SILVA rRNA database (https://www.arb-silva.de) and the Greengenes database (http://greengenes.secondgenome.com). The reference genomes of *Clostridium* and *Prevotella* strains are available in the Human Microbiome Project (https://www.hmpdacc.org). Source data are provided with this paper.

## Code availability

The code scripts used for data processing, analysis, and visualization have been deposited to Zenodo under https://doi.org/10.5281/zenodo.4925167.

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

## Acknowledgements

This work was supported by grants from the Beijing Natural Science Foundation (JQ18020) and the National Natural Science Foundation of China (32025009, 31722031, 31670119, 31870107, 32070122). We thank Lina Hou, Xuehan Li, Ming He, Shuai Chen, Ning Wang, and Zhenqiang Zuo for their assistance in sample preprocessing.

## Author contributions

F.Z. conceived and supervised the study. F.Z. and J.W. designed the study, interpreted the results, and wrote the manuscript. Z.J., X.C., L.Y., and J.Y. collected the samples and clinical information. J.W. and Z.L. processed the samples, prepared the DNA for sequencing, and conducted the experiments. F.Z., J.W., Z.L., X.M., L.D., L.X., B.Z., and H.F. analyzed the data and prepared figures and tables.

## Competing interests

The authors declare no competing interests.
