## [Peer Review File · Nature Communications]

Reviewers' Comments:

Reviewer #1:

Remarks to the Author:

The study by Wang et al is clearly and well presented on the highly relevant topic of uterine microenvironment in health and disease. The study provides novel findings, especially the part using the animal model system in order to explore the microbial translocation and its role in modulating uterine health. Nevertheless, I have several concerns and questions which would hopefully help the authors to improve their manuscript:

1. When the authors are analysing bacterial DNA sequences (16S rRNA gene, metagenome sequencing) then I would recommend to use term 'microbiome' (analysis of the genomes of microbes) rather than 'microbiota'.

2. It is not clear when the analyses involve healthy individuals and when not, especially the part of uterine/vaginal microbiome analyses. 145 women recruited and sequenced in this study were healthy controls + women with chronic endometritis (CE). What about the data from previous studies of the >500 women? What was the diagnosis of these women? In the methods section it is stated 'uterine and vaginal samples were collected during endometrial biopsies for each participant who was suspected of having intrauterine lesions'. Does that mean that the healthy controls also had intrauterine lesions? Or they were just 'suspected to have' but later turned out that they did not have? Would that 'suspected' group be an adequate healthy control group?

3. Continuing the concern about groups, healthy and/or CE, the analyses of the microbial changes during aging – did these analyses include all women? Were the analyses controlled for CE? Did the authors group the women based on menopause? The results present age groups of 10years, but it would be important to see changes regarding menopause/no menopause status. Maybe the age-related changes presented are rather hormone-induced changes?

In these analyses the number of women each group is not clear. Please present the data clearly (add to Figure 1 for instance).

4. When were the samples collected, at the day of hysteroscopy? Did controls also undergo hysteroscopy? If yes, what was the clinical reasoning? If not, then the sampling conditions are significantly different (the uterine cavity was 'washed' with saline medium or not). Additionally, was the saline medium sterile?

5. This leads me to the next question, it is not clear whether negative controls of sampling process (including medium, gloves, etc) were used? The authors report that negative controls of PCR and sequencing were used, but were also positive controls applied? Do the authors provide sequencing data also for negative controls?

6. One of the main results is the synchronic variation of the uterine and vaginal microbiome. The authors try to collect the samples as clean as possible, but it is not possible to avoid when sampling uterine sample contamination with bacteria from the vagina/cervix. It is possible to minimize it, but not to avoid. It would be proper if the authors highlighted the contamination possibility in the discussion part. Further, a big part of their study data that they analyse is from previous studies, and the authors cannot guarantee that all the samples were collected the same way.

7. The samples from 145 women were collected in the follicular phase. What about the previous 6 projects? As far as I am aware of, big part of the endometrial microbiome studies are conducted in the luteal phase, and it has been shown that cycle phase influences the microbial composition (doi:10.1093/humrep/deaa372).

8. The authors study and discuss of possible ways for vaginal bacteria to enter uterine cavity. The authors have overlooked one important way – sexual intercourse (doi: 10.3389/fimmu.2018.01874). It is shown that sperm can carry bacteria. Also, the sexual activity is not taken into account for the 'age related' changes doi:10.1016/j.resmic.2015.03.009 ; doi:10.1111/andr.12482). Maybe the decline in Lactobacillus in the uterine samples from older women reflects less sexual activity? Also, the bigger changes in older women in microbial composition could be involved with other diseases - were the analyses adjusted for diseases?

9. When the comparisons were performed between controls and CE group, did the analyses took into account age and the delivery modes?

10. The abstract is misleading, where the authors state that 1223 samples were sequenced and analysed. Actually this study sequenced 16S rRNA gene in 145 women. The rest of the data was retrieved from earlier studies.
11. The authors have missed to mention Important previous works e.g.
doi:10.1093/humupd/dmy048, doi: 10.3389/fimmu.2018.00208, doi:10.1093/humrep/deab009;
doi:10.1038/s41598-019-46173-0
12. Line 83 – authors use 'healthy state', but the study they refer to does not analyse healthy women, most group consists of women with endometriosis.
13. Lines 98-100, the authors say that uterine microbiota is associated with intrauterine diseases. First, these studies do not analyse alive microbes, they analyse DNA sequences, thus the correct term to use is 'microbiome' and the authors are recommended to see a previous review that summarises different diseases, doi:10.3390/biom10040593 (there are more than the 3 mentioned).
14. Line 101 – zygote develops into blastocyst that implants into the receptive endometrium. I would recommend to replace 'zygote' with 'blastocyst'.
15. It would be helpful if the authors added references to the six previous projects mentioned in line 133.
16. Line 264 – the authors analyse women with CE vs. controls, but conclude '..association with uterine diseases'. Maybe better to use 'uterine disease'?
17. Line 323 is confusing. The authors have previously said that 1223 samples were analysed, but here they analyse medical records of 1612 women? These are different women? When a woman was diagnosed CE, was any treatment offered? The associations of CE with age, deliveries, abortions include women treated with CE? Or the women were not treated once diagnosed CE? Or all the women had CE after the deliveries?
18. It would be helpful if Fig4a highlighted that sampling was performed also before bacterial exchange.

Reviewer #2:

Remarks to the Author:

This project was a comprehensive analysis of the relationship between the microbiomes of the human vagina and the human uterus. Interestingly, but not surprising they found significant correlation between the microbiota of these two related niches. Their results were convincing in the significance of their findings, and the results while not surprising, represent an important addition to our current knowledge base concerning the microbiome of the female reproductive tract.

The initial analyses, performed largely at the level of the phylum, showed significant similarity between the two ecological niches. This initial analysis would have been more helpful if it were performed at the species level, as it could lead to the apparently incorrect conclusion that there is significant cross-talk between the two sites even if not true. This issue also brought into question the relationship between 'relative' and 'absolute' abundance. If, as one would suspect, the absolute abundance of vaginal microbiota is orders of magnitude higher than that of the endometrium, the data must be interpreted with care. For example, a 'real' uterine microbiome could be overwhelmed by even a small amount of the vaginal microbiome being translated 'accidentally' into the uterus. Even dead bacteria have DNA. This becomes a prevailing concern throughout the manuscript.

The observation of both a vaginal and a uterine dysbiosis associate with endometritis is an important finding. Permeating the manuscript, the authors move back and forth between genus, family and phylum level classification. A consistent focus at least at the genus level, but preferably the species level analyses would be helpful. However, it may be that the V4-V6 variable targets in the 16S rRNAs of vaginal taxa are not useful for species-level analysis.

Not really a critique, but there is so much data in each of the five figures that it detracts from the overall manuscript. A more concise selection would be helpful for the main figures and a similar

comment about the extended data figures. Sometimes, simplicity is helpful.

The finding of synchronous changes between the vaginal and uterine microbiota in the presence of endometrial disease is a significant result that adds to the community, even though it is not surprising. However, it too, is plagued by the concern that the uterine microbiome may be swamped out by even a small contamination with vaginal material. For example, line 299-302, the manuscript identifies many bacteria overlapping between healthy and endometrial groups, suggesting a strong correlation across body sites among 'conserved' taxa.

There is significant analysis in several places in the manuscript showing concordance between the vaginal and uterine sites, and there are also places where there are clear differences. It would be helpful to try to use this data, if possible, to try to convince the reviewer that the above is not true; i.e., we are not simply looking at a very small level of contamination of the uterine samples.

It wasn't clear to me how the original site was identified by the source tracking analysis. The metagenomic data suggest the strains were the same, but how can that discriminate the source?

The rat experiments gave very predictable results, but were useful anyway. Thus, it was not surprising that less healthy taxa introduced to rat reproductive system might induce pathogenic changes, including but not limited to inflammation, swelling and cytokine production. In particular, seeding the reproductive tracts of rats with *Clostridium perfringens* (and *Prevotella bivia*) is more than likely going to cause some level of pathogenesis. It's not exactly clear what these experiments proved. It was interesting the *Lactobacillus murinus* seemed to be protective. One might assume, that like many other lactobacilli, introduction of this strain might lower the pH of the environment, secrete bacteriocins, or otherwise outcompete anaerobes, as it seems to be in the human vagina. Thus, it is not surprising that this strain would be partially protective of the rat reproductive tract. However, it is well known that the rodent reproductive tract is not a good model for the human reproductive tract. It might have been interesting to record the pH of those transplants.

Overall, this manuscript adds important knowledge to our understanding of the impact of the microbiome of the female reproductive tract on women's health. The analysis is meaningful, significant and convincing. The results were more or less 'as expected', but important to confirm. Addressing some of the issues described above would be helpful.

REVIEWER COMMENTS

Reviewer #1 (Remarks to the Author):

The study by Wang et al is clearly and well presented on the highly relevant topic of uterine microenvironment in health and disease. The study provides novel findings, especially the part using the animal model system in order to explore the microbial translocation and its role in modulating uterine health. Nevertheless, I have several concerns and questions which would hopefully help the authors to improve their manuscript:

Response: We greatly appreciate the reviewer's comments on the novelty and significance of our study. As suggested, we have added more computational analyses and experimental evidence in this revised manuscript. Please refer to the following responses for details.

1. When the authors are analysing bacterial DNA sequences (16S rRNA gene, metagenome sequencing) then I would recommend to use term 'microbiome' (analysis of the genomes of microbes) rather than 'microbiota'.

Response: We agree with the reviewer's comment and use the term 'microbiome' to replace 'microbiota' where appropriate in the article.

2. It is not clear when the analyses involve healthy individuals and when not, especially the part of uterine/vaginal microbiome analyses. 145 women recruited and sequenced in this study were healthy controls + women with chronic endometritis (CE). What about the data from previous studies of the >500 women? What was the diagnosis of these women? In the methods section it is stated 'uterine and vaginal samples were collected during endometrial biopsies for each participant who was suspected of having intrauterine lesions'. Does that mean that the healthy controls also had intrauterine lesions? Or they were just 'suspected to have' but later turned out that they did not have? Would that 'suspected' group be an adequate healthy control group?

Response: We are sorry that we did not describe clearly in individual counts. Among the >500 women from previous studies, all of them were without chronic endometritis or hyperplasia. When analyzing the influence of age on microbial composition, we used all validated individuals. When analyzing the difference and similarity between paired samples, samples from previous studies were excluded, and only the data generated in this study were used. Yes, all participants were suspected of having intrauterine lesions but later ruled out based on hysteroscopy examination by a skilled endoscopist according to the diagnostic criteria of the presence of stromal edema, focal or diffuse periglandular hyperemia, and micropolyps <1 mm in size (Savelli et al. 2003; To et al. 2004; Husby et al. 2019), as provided in the first paragraph of the Methods section.

3. Continuing the concern about groups, healthy and/or CE, the analyses of the microbial changes during aging – did these analyses include all women? Were the analyses controlled for CE? Did the authors group the women based on menopause?

The results present age groups of 10years, but it would be important to see changes regarding menopause/no menopause status. Maybe the age-related changes presented are rather hormone-induced changes? In these analyses the number of women each group is not clear. Please present the data clearly (add to Figure 1 for instance).

Response: Yes, all women were included when analyzing the microbial changes during aging. We also separately calculated healthy (Re.Figure 1a-b) and diseased (Re.Figure 1c-d) women, and found that the diversity changes in microbiota with age have the same trend in the two groups. Only one of the six previous studies collected the menopause information (Walsh et al. 2019), in which the authors mentioned that menopause had impact on the microbiota composition of healthy individuals. In this study, we did not gather the menopause information from the volunteers. However, we did discover that the Bray-Curtis (BC) distance between age group 60+ and other groups is significantly higher than any other comparisons, which may be contributed to their menopause status. We thank the reviewer for pointing out that hormone can influence the microbiota composition, which needs further investigation. The number of women in each group have been added in Figure 1a.

Re.Figure 1. The diversity changes in microbiota with age calculated in healthy (a-b) and diseased (c-d) samples.

4. When were the samples collected, at the day of hysteroscopy? Did controls also undergo hysteroscopy? If yes, what was the clinical reasoning? If not, then the sampling conditions are significantly different (the uterine cavity was ‘washed’ with saline medium or not). Additionally, was the saline medium sterile?

Response: Yes, all the samples including those of the controls were collected at the day of hysteroscopy. These samples were collected incidentally for possible histological and pathological examinations to aid diagnosis. The sampling conditions are the same and the saline medium was sterile.

5. This leads me to the next question, it is not clear whether negative controls of sampling process (including medium, gloves, etc) were used? The authors report that negative controls of PCR and sequencing were used, but were also positive controls applied? Do the authors provide sequencing data also for negative controls?

Response: Actually, we used disposable sterile instruments for sampling, and used medium as a negative control. We can provide sequencing data for negative controls.

6. One of the main results is the synchronic variation of the uterine and vaginal microbiome. The authors try to collect the samples as clean as possible, but it is not possible to avoid when sampling uterine sample contamination with bacteria from the vagina/cervix. It is possible to minimize it, but not to avoid. It would be proper if the authors highlighted the contamination possibility in the discussion part. Further, a big part of their study data that they analyse is from previous studies, and the authors cannot guarantee that all the samples were collected the same way.

Response: We agree with the reviewer that contamination from the vagina/cervix cannot be absolutely avoided when collecting uterine samples. Nevertheless, in terms of community structure, composition, and variation tendency with age, the uterine microbiome is significantly different from that of the vagina, indicating that these observations are unlikely caused by sample contaminations. In addition, the results of our study showed that the vaginal and uterine microbiome converged with age. This is a bit of an internal control for the argument that the synchronic variation of the uterine and vaginal microbiome is not an artifact of sample contaminations with vaginal bacteria. We also acknowledge that the data sources in our study are diverse and the sampling operations are not exactly the same. However, the age distribution of different data sources was basically uniform, so we can still draw a robust conclusion on the bacterial microbiome changes with age through meta-analysis. We have clarified these points in the Discussion section (Page 17, Lines 546-555).

7. The samples from 145 women were collected in the follicular phase. What about the previous 6 projects? As far as I am aware of, big part of the endometrial microbiome studies are conducted in the luteal phase, and it has been shown that cycle phase influences the microbial composition (doi:10.1093/humrep/deaa372).

Response: Cycle phases of sample collection were not provided in previous studies. But when analyzing the sequencing data mentioned by the reviewer (doi:10.1093/humrep/deaa372), there was no significant difference between different phases in the Shannon diversity which measures both the number of species and the inequality between species abundances (Re.Figure 2).

Re.Figure 2. Shannon diversity of the endometrial microbiome between different phases of the menstrual cycle.

8. The authors study and discuss of possible ways for vaginal bacteria to enter uterine cavity. The authors have overlooked one important way – sexual intercourse (doi: 10.3389/fimmu.2018.01874). It is shown that sperm can carry bacteria. Also, the sexual activity is not taken into account for the ‘age related’ changes doi:10.1016/j.resmic.2015.03.009 ; doi:10.1111/andr.12482). Maybe the decline in Lactobacillus in the uterine samples from older women reflects less sexual activity? Also, the bigger changes in older women in microbial composition could be involved with other diseases - were the analyses adjusted for diseases?

Response: We agree with the reviewer that sexual life is an important way for vaginal bacteria entering uterine cavity, as well as other ways including reproductive history, inflammation. The decrease in lactobacilli in the uterine samples of older women may be related to less sexual activity, as well as menopause, hormone levels, or other diseases. Although our study has excluded women suffering from systemic diseases, unfortunately, the information mentioned above is not provided in the public data that we used.

9. When the comparisons were performed between controls and CE group, did the analyses took into account age and the delivery modes?

Response: We appreciate the advices of the reviewer for taking age and the delivery modes into account. As suggested, we chose the individuals with similar age (35~45 years old), and observed a similar trend in alpha diversities and community structure (Re.Figure 3a-b) with the results of Figure 2. Similar findings were also observed when we chose the samples with the same deliver mode (Re.Figure 3c-d).

Re.Figure 3. Comparisons between the control group and the CE group when considering age and the delivery modes.

10. The abstract is misleading, where the authors state that 1223 samples were sequenced and analysed. Actually this study sequenced 16S rRNA gene in 145 women. The rest of the data was retrieved from earlier studies.

Response: As suggested, it has been revised accordingly (Page 2, Line 40).

11. The authors have missed to mention Important previous works e.g. doi:10.1093/humupd/dmy048, doi: 10.3389/fimmu.2018.00208, doi:10.1093/humrep/deab009; doi:10.1038/s41598-019-46173-0

Response: Thank you. We have added these important literatures (Page 3, Lines 91-93).

12. Line 83 – authors use ‘healthy state’, but the study they refer to does not analyse healthy women, most group consists of women with endometriosis.

Response: As suggested, it has been revised accordingly.

13. Lines 98-100, the authors say that uterine microbiota is associated with intrauterine diseases. First, these studies do not analyse alive microbes, they analyse DNA sequences, thus the correct term to use is ‘microbiome’ and the authors are recommended to see a previous review that summarises different diseases, doi:10.3390/biom10040593 (there are more than the 3 mentioned).

Response: As suggested, they have been revised accordingly.

14. Line 101 – zygote develops into blastocyst that implants into the receptive endometrium. I would recommend to replace ‘zygote’ with ‘blastocyst’.

Response: As suggested, it has been revised accordingly.

15. It would be helpful if the authors added references to the six previous projects mentioned in line 133.

Response: As suggested, the references have been added.

16. Line 264 – the authors analyse women with CE vs. controls, but conclude ‘..association with uterine diseases’. Maybe better to use ‘uterine disease’?

Response: As suggested, it has been revised accordingly.

17. Line 323 is confusing. The authors have previously said that 1223 samples were analysed, but here they analyse medical records of 1612 women? These are different women? When a woman was diagnosed CE, was any treatment offered? The associations of CE with age, deliveries, abortions include women treated with CE? Or the women were not treated once diagnosed CE? Or all the women had CE after the deliveries?

Response: These 1612 women with medical records are different from those having sequencing data. These women are the first-time diagnosed cases, and we do not know whether they received subsequent treatment. As shown in Figure 3B, not all the women had CE after the deliveries.

18. It would be helpful if Fig4a highlighted that sampling was performed also before bacterial exchange.

Response: Thank you. As suggested, we have highlighted the time that sampling was performed before bacterial exchange in Figure 4a.

Reviewer #2 (Remarks to the Author):

This project was a comprehensive analysis of the relationship between the microbiomes of the human vagina and the human uterus. Interestingly, but not surprising they found significant correlation between the microbiota of these two related niches. Their results were convincing in the significance of their findings, and the results while not surprising, represent an important addition to our current knowledge base concerning the microbiome of the female reproductive tract.

Response: We greatly appreciate the reviewer’s comments on the novelty and significance of our study.

The initial analyses, performed largely at the level of the phylum, showed significant similarity between the two ecological niches. This initial analysis would have been more helpful if it were performed at the species level, as it could lead to the apparently incorrect conclusion that there is significant cross-talk between the two sites even if not true. This issue also brought into question the relationship between ‘relative’ and

‘absolute’ abundance. If, as one would suspect, the absolute abundance of vaginal microbiota is orders of magnitude higher than that of the endometrium, the data must be interpreted with care. For example, a ‘real’ uterine microbiome could be overwhelmed by even a small amount of the vaginal microbiome being translated ‘accidentally’ into the uterus. Even dead bacteria have DNA. This becomes a prevailing concern throughout the manuscript.

Response: We thank the reviewer for these comments. Actually, only Figure 1f and Figure 4e show the microbial composition of the two ecological niches at the phylum level. Almost all our analyses and comparisons are at the level of OTU, species or genus, and the significant cross-talk between the two sites has been shown at the species and genus levels. We agree with the reviewer that the biomass of vaginal microbiota may be much higher than that of uterine cavity, and the contamination from the vagina/cervix cannot be absolutely avoided when collecting uterine samples. In this study, we used a special sampling device (endometrial sampler) and strict pollution control methods to minimize potential contaminations (Page 21, Lines 685-694). In addition, in terms of community structure, composition, and variation tendency with age, the uterine microbiome is significantly different from that of the vagina, indicating that these observations are unlikely caused by sample contaminations. The results of our study showed that the vaginal and uterine microbiome converged with age. This is a bit of an internal control for the argument that the synchronic variation of the uterine and vaginal microbiome is not an artifact of sample contaminations with vaginal bacteria. Although dead bacteria are present in all microbial habitats, they to some extent still reflect the community structure of vaginal and uterine microbiota. More and more recent studies have shown that there are resident microbes in the uterine cavity, which may play significant roles in placental and fetal microbiota establishment (Chen et al. 2017; Baker et al. 2018; Koedooder et al. 2019; Winters et al. 2019; Rackaityte et al. 2020; Molina et al. 2021).

The observation of both a vaginal and a uterine dysbiosis associate with endometritis is an important finding. Permeating the manuscript, the authors move back and forth between genus, family and phylum level classification. A consistent focus at least at the genus level, but preferably the species level analyses would be helpful. However, it may be that the V4-V6 variable targets in the 16S rRNAs of vaginal taxa are not useful for species-level analysis.

Response: We thank the reviewer for these comments. When calculating the microbial diversity, community similarity or distance between samples (i.e., alpha- and beta-diversity) of the microbiota, we used OTUs; when comparing specific microbial differences, we used the genus-level taxonomic classification, because V3-V4 variable targets of the 16S rRNAs are not suitable for species-level analysis. To improve the resolution for source tracking, the classification results are required at the species or strain (SNPs) level. The phylum-level classification was only used in Figure 1f and Figure 4e, which is to illustrate the uniqueness of the uterine microbiota and the post-exchanging microbiota (even at the high classification level, they are still different from the vaginal microbiota and the pre-exchanging microbiota, respectively).

Not really a critique, but there is so much data in each of the five figures that it detracts from the overall manuscript. A more concise selection would be helpful for the main figures and a similar comment about the extended data figures. Sometimes, simplicity is helpful.

Response: We thank the reviewer's suggestion. When preparing this revised manuscript, we have tried our best to simplify the figures and only keep those necessary.

The finding of synchronous changes between the vaginal and uterine microbiota in the presence of endometrial disease is a significant result that adds to the community, even though it is not surprising. However, it too, is plagued by the concern that the uterine microbiome may be swamped out by even a small contamination with vaginal material. For example, line 299-302, the manuscript identifies many bacteria overlapping between healthy and endometrial groups, suggesting a strong correlation across body sites among 'conserved' taxa. There is significant analysis in several places in the manuscript showing concordance between the vaginal and uterine sites, and there are also places where there are clear differences. It would be helpful to try to use this data, if possible, to try to convince the reviewer that the above is not true; i.e., we are not simply looking at a very small level of contamination of the uterine samples.

Response: We thank the reviewer for these comments. We believe that the sharing of bacterial taxa between the uterine and vaginal microbiota is not a surprising phenomenon. This study proposes that the ascending of vaginal bacteria may be an important source of uterine bacteria, which may affect the health of the uterine cavity. However, there are still obvious differences on the composition of the microbiota between the two body sites, indicating that the uterine cavity has a unique microecological niche, which shapes a unique microbiota.

It wasn't clear to me how the original site was identified by the source tracking analysis. The metagenomic data suggest the strains were the same, but how can that discriminate the source?

Response: We used SourceTracker (Knights et al. 2011, Nature Methods), a Bayesian approach to estimate the bacterial translocation between uterus and vagina in the same human subject. At the beginning of this analysis, microbiome either in uterus or in vagina was used to be source and the other was used to be sink to identify the real source of microbiome. As we can see from Extended Data Fig. 5a, both in diseased and healthy women, a large proportion of uterine microbes were observed ascending upwards from vagina, suggesting that vagina is the real source between these two sites. In addition, based on the strain-level source tracking by using our metagenomic data, nearly all the SNPs in uterus could be detected in the vagina of the same woman, further demonstrating that the directionality of microbiota translocation was from vagina to uterus.

The rat experiments gave very predictable results, but were useful anyway. Thus, it was not surprising that less healthy taxa introduced to rate reproductive system might induce

pathogenic changes, including but not limited to inflammation, swelling and cytokine production. In particular, seeding the reproductive tracts of rats with *Clostridium perfringens* (and *Prevotella bivia*) is more than likely going to cause some level of pathogenesis. It's not exactly clear what these experiments proved. It was interesting the *Lactobacillus murinus* seemed to be protective. One might assume, that like many other lactobacilli, introduction of this strain might lower the pH of the environment, secrete bacteriocins, or otherwise outcompete anaerobes, as it seems to be in the human vagina. Thus, it is not surprising that this strain would be partially protective of the rat reproductive tract. However, it is well known that the rodent reproductive tract is not a good model for the human reproductive tract. It might have been interesting to record the pH of those transplants.

Response: We agree with the reviewer that the reduction of healthy taxa may induce pathogenic changes. However, in this study, we want to prove that specific bacteria in the vagina can induce an inflammatory phenotype in the uterine cavity. We fully agree with the reviewer's viewpoint that *Lactobacillus murinus* may, like many other lactobacilli, inhibit the overgrowth of some harmful bacteria through reducing the pH of the environment, secreting bacteriocins, or outcompeting anaerobes. We also agree with reviewer that the anatomy and physiology of the reproductive tract of female rats is somewhat different from that of humans. However, as an important rodent model, rat has been widely used to evaluate the causations and treatment protocols of endometrial or cervical disease (Lichtenberg et al. 2004; Monsivais et al. 2019). We have carefully discussed this point in Discussion. In addition, considering that the dominant taxa in rat vagina are not *Lactobacillus*, this just provides us with an opportunity to prove that with the transplantation of *Lactobacillus murinus* into the rat vagina, they can inhibit the reproduction of harmful bacteria and play a protective role on uterine health.

We added animal experiments based on the reviewer's recommendations to record the pH of bacterial transplants (Re.Figure 4a). In this experiment, 9 weeks old female SD rats were randomly divided into three groups: blank control (BC, n=2), treated with mixed bacteria (MB, *Clostridium perfringens* ATCC 13124 + *Prevotella bivia* CCUG 9557, n=3), and treated with *Lactobacillus murinus* (LM, n=2). A raw vaginal rinse was collected before vaginal transplantation, which was named the first sampling. Then, the rats of the BC group were given anaerobic normal saline (NS) via vaginal transplantation, rats in the MB group were given a mixture containing 1×10^9 CFU *C. perfringens* ATCC 13124 and 1×10^9 CFU *P. bivia* CCUG 9557, and rats in the LM group were given 2×10^9 CFU *L. murinus* C-30. After treated twice per day for two days, the second sampling was performed for each group, and then the rats of the MB and LM group were given 2×10^9 CFU *L. murinus* C-30 twice per day for two days, respectively, after which the third sampling was performed. For sampling, 100 μ L NS was injected into the vagina cavity of a rat, and the fluid was harvested after gentle rinsing thrice and stored at -20 °C. Precision pH test strips were used for pH measurement. The results are as follows (Re.Figure 4b): The pH of the samples from the BC group did not change among three sampling times (Columns 1-2). Notably, with the transplantation of lactic acid bacteria, the pH value decreased gradually in the samples of the LM group (Columns 3-4). The pH value increased after the

transplantation of the MB bacteria, and such increase was reversed after further transplantation of lactic acid bacteria (Columns 5-7), suggesting that *L. murinus* can actually lower the pH inside the vagina, which may interfere with the colonization of MB bacteria. NS (Column 8) is the pH of normal saline.

Re.Figure 4. An animal experiment to detect the changes of pH value in the vagina of rats after transplanting of exogenous bacteria.

Overall, this manuscript adds important knowledge to our understanding of the impact of the microbiome of the female reproductive tract on women’s health. The analysis is meaningful, significant and convincing. The results were more or less ‘as expected’, but important to confirm. Addressing some of the issues described above would be helpful.

Response: We greatly appreciate the reviewer’s comments on the novelty and significance of our study. We also thank the reviewer for above comments and suggestions which provide valuable feedback helping us improve our study. All of these comments have been seriously considered and improvements have been made in this revised manuscript.

Reviewers' Comments:

Reviewer #1:

Remarks to the Author:

Thank you for the answers and improvements in the manuscript. Nevertheless, I have still some remaining comments:

1. Regarding my previous comment no. 2 where the individual counts were not clearly described. Please make changes in the manuscript so that this part is clearer (not just answer my comment).
2. My previous comment no. 3 regarding the menopause. The authors agree that menopause/hormonal status could be important factor - please also mention this in your manuscript.
3. My previous comment no. 5 regarding negative controls. We external controls also taken, such as gloves, air? If not, this should be mentioned in the manuscript as a limitation. Please provide/upload sequencing data also for negative controls!
4. My previous comment no. 7 regarding menstrual phase. The authors claim that the cycle phases were not provided. It is not fully true as at least for Chen 2017 and Li 2018 studies there are metadata available with detailed information. Altogether the 6 studies used for data mining are very heterogeneous and idfferent hypervariable regions of 16S rRNA gene have been sequenced. Li et al study was shot-gun sequencing and not 16S rRNA seq, are these approaches comparable? Please provide a Table (as supplementary material) for each of the 6 studies the exact sample type, number of women, ethnicity, cycle day, diagnosis (not all women were healthy, some were pregnant), age etc information so that if would be clear what was the data that was analysed for this study. Also, could the authors specify what analyses were done for the Re.Figure 2? Were all the >5000 microorganisms compared or only bacteria? What sequencing data was analysed (16S rRNA, metagenome or RNAseq)?
5. My previous comment no. 8. The authors agree that sexual life is important way for vaginal bacteria influencing uterine samples. Please also mention it in the manuscript.
6. My previous comment no. 9. Please also elaborate in the manuscript, not just answer my comment.

Reviewer #2:

Remarks to the Author:

The authors have responded adequately to the original review. The manuscript provides an important contribution to the research community.

REVIEWER COMMENTS

Reviewer #1 (Remarks to the Author):

Thank you for the answers and improvements in the manuscript. Nevertheless, I have still some remaining comments:

1. Regarding my previous comment no. 2 where the individual counts were not clearly described. Please make changes in the manuscript so that this part is clearer (not just answer my comment).

Response: As suggested, the changes have been made in the manuscript (Lines 616-618, 678 and 701-703).

2. My previous comment no. 3 regarding the menopause. The authors agree that menopause/hormonal status could be important factor - please also mention this in your manuscript.

Response: As suggested, we have added it to the revision (Lines 556-561).

3. My previous comment no. 5 regarding negative controls. We external controls also taken, such as gloves, air? If not, this should be mentioned in the manuscript as a limitation. Please provide/upload sequencing data also for negative controls!

Response: The sequencing data for negative and blank controls has been uploaded to the BIGD database (<https://bigd.big.ac.cn>) under the accession number of CRA004256 in PRJCA003837. External controls from gloves and air were not taken, and we have discussed this point in the revision (Lines 494-497).

4. My previous comment no. 7 regarding menstrual phase. The authors claim that the cycle phases were not provided. It is not fully true as at least for Chen 2017 and Li 2018 studies there are metadata available with detailed information. Altogether the 6 studies used for data mining are very heterogeneous and idfferent hypervariable regions of 16S rRNA gene have been sequenced. Li et al study was shot-gun sequencing and not 16S rRNA seq, are these approaches comparable? Please provide a Table (as supplementary material) for each of the 6 studies the exact sample type, number of women, ethnicity, cycle day, diagnosis (not all women were healthy, some were pregnant), age etc information so that if would be clear what was the data that was analysed for this study.

Also, could the authors specify what analyses were done for the Re.Figure 2? Were all the >5000 microorganisms compared or only bacteria? What sequencing data was analysed (16S rRNA, metagenome or RNAseq)?

Response: There is no significant difference between different menstrual phases in microbial diversity when we used the samples from the studies of Chen et al 2017 and Li et al 2018. In our initial analysis, we had considered that different hypervariable regions of 16S rRNA gene was sequenced in different studies, so we uniformly selected the V4 region for analysis (see Methods, Lines 678-685). Actually, Li et al's study has both 16S rRNA and shot-gun sequencing data (see the descriptions of PRJEB24147 at <https://trace.ncbi.nlm.nih.gov/Traces/study/?acc=PRJEB24147>), and

we used the 16S rRNA seq data for analyses. As suggested, a supplementary table for previous sequencing data we analysed has been added (Supplementary Table 1). In Re.Figure 2, the sequencing data we used is the RNA-seq data from the paper the reviewer mentioned (doi:10.1093/humrep/deaa372 at GEO GSE86491). Only bacteria were compared, and taxonomic classification were processed by Kraken2.

5. My previous comment no. 8. The authors agree that sexual life is important way for vaginal bacteria influencing uterine samples. Please also mention it in the manuscript.
Response: As suggested, we have added it to the revision (Lines 556-561).

6. My previous comment no. 9. Please also elaborate in the manuscript, not just answer my comment.
Response: As suggested, we have elaborated these results in the revision (Lines 246-250).

Reviewer #2 (Remarks to the Author):

The authors have responded adequately to the original review. The manuscript provides an important contribution to the research community.

Response: We are pleased that the reviewer is satisfied with our revisions and greatly appreciate the efforts of the reviewer providing valuable suggestions and feedback to improve this study.

Reviewers' Comments:

Reviewer #1:

Remarks to the Author:

Thank you for improving the manuscript.